# Precise CAG repeat contraction in a Huntington's Disease mouse model is enabled by gene editing with SpCas9-NG

Seiya Oura [1,2,11], Taichi Noda [1,3,11], Naoko Morimura[4], Seiji Hitoshi[4], Hiroshi Nishimasu[5,6], Yoshitaka Nagai[7,8,9], Osamu Nureki[5] & Masahito Ikawa [1,2,10 ✉]

The clustered regularly interspaced palindromic repeats (CRISPR)/Cas9 system is a research hotspot in gene therapy. However, the widely used *Streptococcus pyogenes* Cas9 (WT-SpCas9) requires an NGG protospacer adjacent motif (PAM) for target recognition, thereby restricting targetable disease mutations. To address this issue, we recently reported an engineered SpCas9 nuclease variant (SpCas9-NG) recognizing NGN PAMs. Here, as a feasibility study, we report SpCas9-NG-mediated repair of the abnormally expanded CAG repeat tract in Huntington's disease (HD). By targeting the boundary of CAG repeats with SpCas9-NG, we precisely contracted the repeat tracts in HD-mouse-derived embryonic stem (ES) cells. Further, we confirmed the recovery of phenotypic abnormalities in differentiated neurons and animals produced from repaired ES cells. Our study shows that SpCas9-NG can be a powerful tool for repairing abnormally expanded CAG repeats as well as other disease mutations that are difficult to access with WT-SpCas9.

[1] Department of Experimental Genome Research, Research Institute for Microbial Diseases, Osaka University, Osaka, Japan. [2] Graduate School of Pharmaceutical Sciences, Osaka University, Osaka, Japan. [3] Division of Reproductive Biology, Institute of Resource Development and Analysis, Kumamoto University, Kumamoto, Japan. [4] Department of Integrative Physiology, Shiga University of Medical Science, Otsu, Shiga, Japan. [5] Department of Biological Sciences, Graduate School of Science, The University of Tokyo, Tokyo, Japan. [6] Department of Structural Biology, Research center for Advanced Science and Technology, The University of Tokyo, Tokyo, Japan. [7] Department of Degenerative Neurological Diseases, National Institute of Neuroscience, National Center of Neurology and Psychiatry, Kodaira, Japan. [8] Department of Neurotherapeutics, Osaka University Graduate School of Medicine, Osaka, Japan. [9] Department of Neurology, Kindai University Faculty of Medicine, Osaka, Japan. [10] Laboratory of Reproductive Systems Biology, Institute of Medical Science, The University of Tokyo, Tokyo, Japan. [11] These authors contributed equally: Seiya Oura, Taichi Noda. ✉email: ikawa@biken.osaka-u.ac.jp

Clustered regularly interspaced short palindromic repeats (CRISPR)/CRISPR-associated 9 (Cas9) is an RNA-guided genome editing tool[1–4], originating from an adaptive immune system in bacteria and archaea[5]. Using a guide RNA (gRNA) consisting of a CRISPR RNA (crRNA) and a *trans*-activating crRNA (tracrRNA), Cas9 nuclease recognizes its target DNA adjacent to a protospacer adjacent motif (PAM) and induces double-strand breaks at the locus. Owing to its simplicity and efficient nature, CRISPR/Cas9 has revolutionized gene-manipulation researches and its application, including gene therapy. However, the widely used Cas9 from *Streptococcus pyogenes* (SpCas9) requires NGG PAM for its target recognition[6], thereby restricting targetable genomic loci. To address this issue, we recently reported an engineered SpCas9 nuclease variant (SpCas9-NG) recognizing NGN-PAMs[7], broadening the range of targetable disease mutations.

Trinucleotide repeat disorders are a set of inherited diseases caused by abnormally expanded trinucleotide tracts, leading to loss or gain of function of the gene products. One of the most common trinucleotide sequences implicated in human diseases is CAG appearing in the coding region[8,9], represented by Huntington's disease (HD). As the CAG sequence encodes glutamine (Q), these disordered are referred to as polyQ disorders. A common symptom of polyQ disorders is progressive neural cell degeneration caused by accumulated homopolymeric expansion proteins[10,11]. There is no cure for polyQ disorders as well as other trinucleotide repeat disorders, although disease-modifying therapies focusing on reducing disease-causing gene products have been actively investigated[12].

CRISPR/Cas9 can be a breakthrough for causal treatment by directly targeting and repairing responsible genes. The most straightforward application of CRISPR/Cas9 is to excise the expanded repeat tracts[13–16]. This was accomplished by designing two gRNAs outside the repeat tracts, disrupting the CAG repeats and extra intragenic sequences. An alternative strategy is to integrate exogenous DNA into the loci (homologous recombination; HR) to precisely repair the responsible genes[17]. However, HR efficiency is lower than NHEJ. Thus, there remained a dilemma as CAG repeats are not be recognized by WT-SpCas9.

Here, we report the contraction of HD-related CAG repeat tracts by SpCas9-NG-mediated genome editing. Using SpCas9-NG, we induced double-strand breaks (DSBs) inside the CAG repeat tract, resulting in precise repair of the responsible gene. Our method has the potential to offer an alternative option for repairing expanded repeat sequences.

## Results

**Genome editing efficiency of SpCas9-NG.** Before targeting expanded CAG repeat tracts, we examined the genome editing efficiency with SpCas9-NG in three different cells, HEK293T cells, ES cells, and zygotes. First, we transfected HEK293T cells with gRNA/SpCas9-expressing plasmids and a reporter plasmid[18] containing a portion of *Rosa26*, *Cetn1*, and *Dnajb13*. Thirty-six hours after transfection, we observed green fluorescence appearing after homology-directed repair (HDR) (Fig. 1a). As previously reported[7], SpCas9-NG cleaved target sequences with NGA/NGT/NGC PAMs more efficiently than WT-SpCas9, albeit a bit less efficiently with an NGG PAM (Fig. 1b and Supplementary Fig. 1a–c). In NGA/NGT sites, SpCas9-NG showed higher activity than WT-SpCas9 in six out of six targets. However, in NGC sites, SpCas9-NG showed higher activity than WT-SpCas9 only in two out of six targets, indicating that SpCas9-NG is less active at NGC sites than NGA/NGT sites, consistent with a previous study[7].

Next, we transfected mouse embryonic stem (ES) cells with gRNA/SpCas9-expressing plasmids used in the above assay (Fig. 1c). Consistent with the result obtained with HEK293T cells, the cleavage activity of SpCas9-NG was higher than that of WT-SpCas9 at NGA/NGT/NGC sites, although WT-SpCas9 also cleaved some NGA/NGT/NGC PAM target sequences (Fig. 1d, Supplementary Fig. 1d-e, and Supplementary Data 2). Finally, we asked whether SpCas9-NG cleaves genomic DNA in zygotes. We electroporated gRNA/SpCas9 ribonucleoprotein complexes or microinjected gRNA/SpCas9-expressing plasmid targeting the *Dnajb13* locus into mouse zygotes and then examined indel efficiency using pooled blastocyst samples (Fig. 1e). In electro-poration of ribonucleoprotein complexes, SpCas9-NG showed higher activity than WT-SpCas9 at 2 NGA target sites (Fig. 1f), indicating that SpCas9-NG has the NGA-PAM preference in zygotes, as observed in HEK293T and ES cells. In contrast, when microinjecting gRNA/SpCas9-expressing plasmids were micro-injected into zygotes, SpCas9-NG cleaved the three out of three target sequences more efficiently than WT-SpCas9 (Fig. 1f). These results showed that SpCas9-NG can cleave NGA/NGT/NGC sites in various types of cells.

**CAG repeat contraction.** As a model of triplet nucleotide dis-orders, we focused on the transgenic R6/2 mouse[19] with HD-patient-derived *HTT* exon1 with 140–147 CAG repeats. First, we established ES cells from the R6/2 mouse line (hereafter referred to as R6/2 ES cells) (Fig. 2a). Then, we designed four gRNAs on the boundary of human *HTT* CAG repeat tracts (Fig. 2b), and trans-fected R6/2 ES cells with plasmids expressing SpCas9 and one or two gRNAs (referred to as the one- or two-hit method, respectively). In PCR screening, all combinations in the two-hit method efficiently downsized signals (Supplementary Fig. 2a and 3), although some clones showed upsized band-shift due to further expansion of repeat tracts. By direct sequencing, we confirmed that 10 out of 41 clones had an in-frame CAG deletion (Supplementary Fig. 2b). However, the one-hit method with gRNA-S1 and gRNA-S2 also downsized signal, albeit not with gRNA-AS1 and gRNA-AS2 (Fig. 2c and Supplementary Figs. 4–6). As with the two-hit method, we con-firmed that 21 out of 87 clones resulted in an in-frame CAG deletion (Fig. 2d, e and Supplementary Data 3). gRNA-S2 more efficiently contracted the CAG repeat tract than gRNA-S1 (Fig. 2e, f). To examine one-hit method is efficient enough for CAG repeat con-traction, we targeted the 3 longest endogenous CAG repeat sequences in mouse genomic DNA (**chr7:** 36,559,029–36,559,121 [31 repeats]; chr13: 4,490,789–4,490,884 [32 repeats]; and chr17: 55,547,368–55,547,460 [31 repeats]). We designed gRNAs in the same way with gRNA-S1 and gRNA-AS1, using the 3rd–4th CAG sequence as PAMs (Supplementary Fig. 2c). Consistent with the result of *HTT* CAG repeat, we observed downsized band-shift more efficiently in SpCas9-NG than in WT-SpCas9 (Supplementary Fig. 2d, e, 7, and 8). All gRNA-Ss showed higher targeting efficiency than gRNA-ASs. These results indicate that one gRNAs designed on the boundary are enough to remove long CAG repeat tracts.

**Off-target analysis.** To examine off-target events in human cells, we transfected HEK293T cells with gRNA-S1 and gRNA-S2 expression vectors (Fig. 3a), and examined mutation rate in all candidate sites by PCR-seq. The off-target candidate sites have the exact match of seed sequence (12mer) and more than five other nucleotides. With gRNA-S1, SpCas9-NG efficiently cleaved the on-target site than WT-SpCas9 (Fig. 3b and Supplemental Fig. 9a), while mutation rates at off-target sites were comparable even with mock-transfected groups. In contrast, we detected higher off-target mutations with gRNA-S2, although gRNA-S2/SpCas9-NG was more efficient in the on-target site than gRNA-S1/SpCas9-NG (Fig. 3b

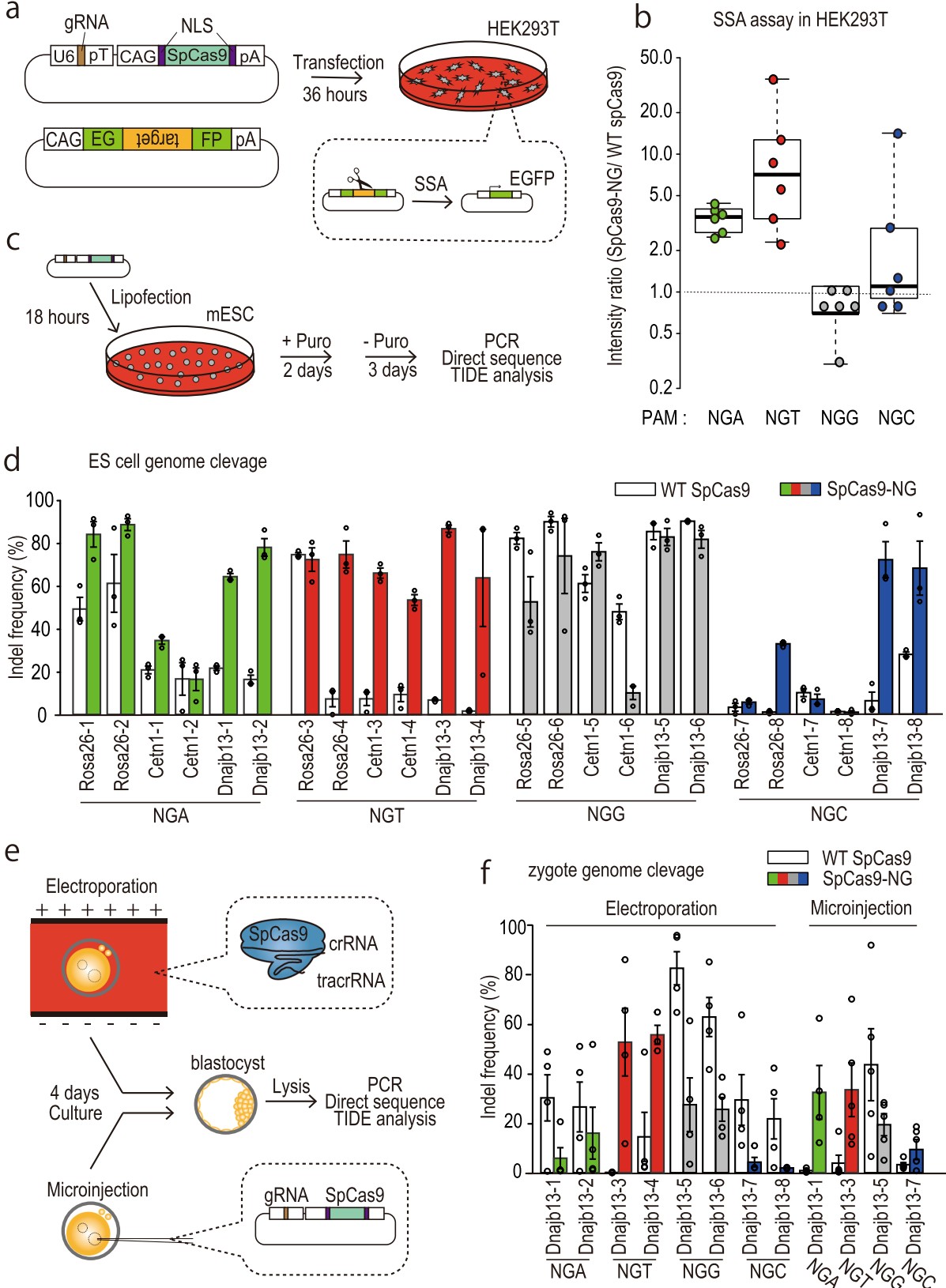

and Supplemental Fig. 9). These results suggest that the gRNA-S1 has a lower risk in human *HTT* targeting.

**Reversal of phenotypic abnormalities in vitro**. We evaluated the phenotype upon CAG repeat contraction. First, we differentiated

original and genome-edited R6/2 ES cells in vitro (Fig. 4a), and then examined neuronal differentiation and aggregation of hun-tingtin protein by anti-βIII tubulin (an early neural marker) and anti-huntingtin staining, respectively. As shown in Fig. 4b, c we observed more neural cells in genome-edited R6/2 embryo bodies (EBs) than in mock-treated R6/2 EBs. Further, genome-edited

**Fig. 1 Genome editing efficiency using SpCas9-NG. a** Strategy of the single-strand annealing (SSA) assay used to evaluate DNA cleavage activity. After transfection of 293 T cells with gRNA/Cas9 and reporter plasmids, the EGFP fluorescence intensity was subsequently measured. **b** Intensity ratio (SpCas9-NG /WT-SpCas9) in the SSA assay. **c** Strategy of ESC genome editing for examining indel efficiency. After the transfection of ES cells with gRNA/Cas9 plasmids targeting *Rosa26*, *Cetn1*, and *Dnajb13* loci, we examined indel efficiency by directly sequencing the PCR amplicons and TIDE software. **d** Indel frequency in pooled ES cells determined by TIDE analysis ($n = 3$, taken from distinct samples). Error bars indicate standard error. **e** Strategy of zygote genome editing for examining indel efficiency. After electroporating gRNA/SpCas9 ribonucleoprotein complexes or microinjecting gRNA/SpCas9 plasmids targeting the *Dnajb13* locus into mouse zygotes, we examined indel efficiency in blastocysts by direct sequencing of the PCR amplicons and TIDE software. **f** Indel frequency in pooled blastocysts was determined by TIDE analysis ($n = 3$; taken from distinct samples). Error bars indicate standard error.

R6/2 EBs exhibited no huntingtin aggregates (Fig. 4b), despite the presence of many neural cells. This result indicates that phenotypic abnormalities were ameliorated in differentiated neurons.

**Phenotypic analysis with chimeric mice.** Next, we produced chimeric mice by injecting two clones of genome-edited (hereafter referred to as s2-11 with 35–36 CAG repeats and s2-21 with 2 CAG repeats, respectively) and original R6/2 ES cells into WT ICR embryos. First, to examine what percentage of cells should be genetically corrected for symptom amelioration, we analyzed chimeric mice with original R6/2 ESCs (Fig. 5a). One of the outstanding phenotypes of R6/2 mice is weight reduction[19] (also see Fig. 6a). Chimeric mice with 20% or more WT cells kept gaining weight, whereas those with less than 10% of WT cells lost weight after 12 weeks of age (Fig. 5b, Table 1, and Supplementary Data 4). This result indicated that around 20–30% of genetically corrected cells could alleviate the symptom, which agrees with the previous chimeric study indicating that 30–70% of WT cell contribution delayed the onset of R6/2 symptom[20]. When we determined the ESC contribution ratio precisely by PCR amplification of a microsatellite D15Mit266[21], the chimeric rate did not differ between several tissues (Fig. 5c–e), suggesting that the expanded CAG repeat did not affect cell viability or proliferation ability in the brain before 12 weeks of age. Next, we examined huntingtin aggregates, another outstanding phenotype, by immunohistochemistry (Fig. 5f). The number of HTT foci (Fig. 5g) showed a tendency of an exponential decrease in association with reducing R6/2 cell ratio (Fig. 5h). When chimeric mice were divided into two groups depending on their chimeric ratio (less than 20% and more than 30% of WT cells), they showed a significant difference (Fig. 5h). This result suggested that CAG repeat contraction in 20–30% of cells dramatically improves the symptom.

**Reversal of phenotypic abnormalities in vivo.** To confirm the CAG repeat contraction reverse the phenotype completely, we mated s2-11 and s2-21 chimeric mice with WT C57BL/6 N mice to produce F1 mice and analyzed their phenotype. Whereas R6/2 mice showed significantly lower weight than WT littermate control after 8 weeks old (Fig. 6a), both s2-11 and s2-21 mice continued to gain weight, comparable to their littermates (Fig. 6b, c). Further, genome-edited mice reached 20 weeks old and kept gaining weight, at the point where no R6/2 mice remained alive (Fig. 6d). As both s2-11 and s2-21 showed recovery in body weight and longevity, we focused on s2-21 for further analysis. Another phenotype of R6/2 mice is dyskinesia of the limbs when suspended by the tail. As shown in Fig. 6e (also see Supplementary Movie 1), s2-21 mice could hold their hind limbs outward to steady themselves, whereas R6/2 mice could not, demonstrating a foot-clasping posture. The constant tremors seen in R6/2 mice were also improved in s2-21 mice (Supplementary Movie 2). Further, when we examined brain histology by HE staining, cerebral atrophy observed in R6/2 mice was improved in s2-21 mice (Fig. 6f–h). Finally, we examined huntingtin

aggregates in the striatum and cortex by immunostaining. No aggregates were observed in s2-21 mouse brains (Fig. 6i–n and Supplementary Fig. 10). These results show that CAG repeat contraction completely recovered phenotypic abnormalities.

**Discussion**

The purpose of the present study was to contract expanded trinucleotide repeats using SpCas9-NG. First, we demonstrated that SpCas9-NG cleaves the ES cell genome at NGA/NGT/NGC sites more efficiently than WT-SpCas9. Then, we precisely contracted CAG repeat tracts in ES cells derived from R6/2 Huntington's disease model mice. Furthermore, we confirmed the complete recovery of phenotypic abnormalities of R6/2 transgenic mice in vivo.

In terms of genome editing activity, SpCas9-NG cleaved NGA/NGT/NGC PAMs efficiently in HEK293T cells, ES cells, and zygotes. As a note, the efficiency of gRNA/SpCas9-NG ribonucleoprotein complex-mediated genome editing was not satisfactory in our hand. Previously, Fujii et al. successfully generated a tyrosinase knockout by microinjecting SpCas9-NG mRNA and gRNA into the cytoplasm of zygotes[22]. The long-expression might be needed for SpCas9-NG due to its reduced cleavage efficiency (see NGG sites of Fig. 1b, d, and f), as we only saw efficient cleavage with plasmid injection. Improvement of SpCas9-NG activity will be required. Recently developed SpG and SpRY Cas9 variants[23] are also worth trying zygote genome editing, as these variants exhibited higher editing activity than SpCas9-NG, at least in cultured cells.

To contract the expanded CAG repeat tract, we originally designed two gRNAs on both the repeat tract boundaries (two-hit). Surprisingly, however, we successfully contracted the repeat tract only by one gRNA with enough efficiency (one-hit), and deleted almost the whole repeat sequence in some clones. This large deletion occurred, probably because CAG repeat tracts become unstable slipped-strand structures upon double-strand breaks[24,25]. This unstable nature might also explain the further expansion of repeat tracts upon SpCas9-NG introduction. Another consideration for this large deletion is repetitive cleavage of CAG repeat tracts as homology-mediated repair (single-strand annealing or microhomology-mediated repair) results in the reappearance of the gRNA-recognizing sequence. We could not evaluate the two-hit method in this study because gRNA-AS1 and gRNA-AS2 could not guide Cas9 to the target sequence efficiently (Fig. 2c and 2e). More trials on different targets with different gRNA sets will be needed to conclude which method is better.

As the off-target analysis shows, our strategy is a trade-off between efficient CAG repeat contraction and low off-target cleavages. Although genome-wide off-target analysis[26–28] in human neural ES/iPS cells is ideal, our result showed that gRNA-S1 has less off-target risk than gRNA-S2. High fidelity mutations[29–35] are needed to be introduced into SpCas9-NG to minimize the risk. It should be noted that the use of Cas9 nickase (D10A or N863A mutations) also has been reported to reduce off-target event[36–38].

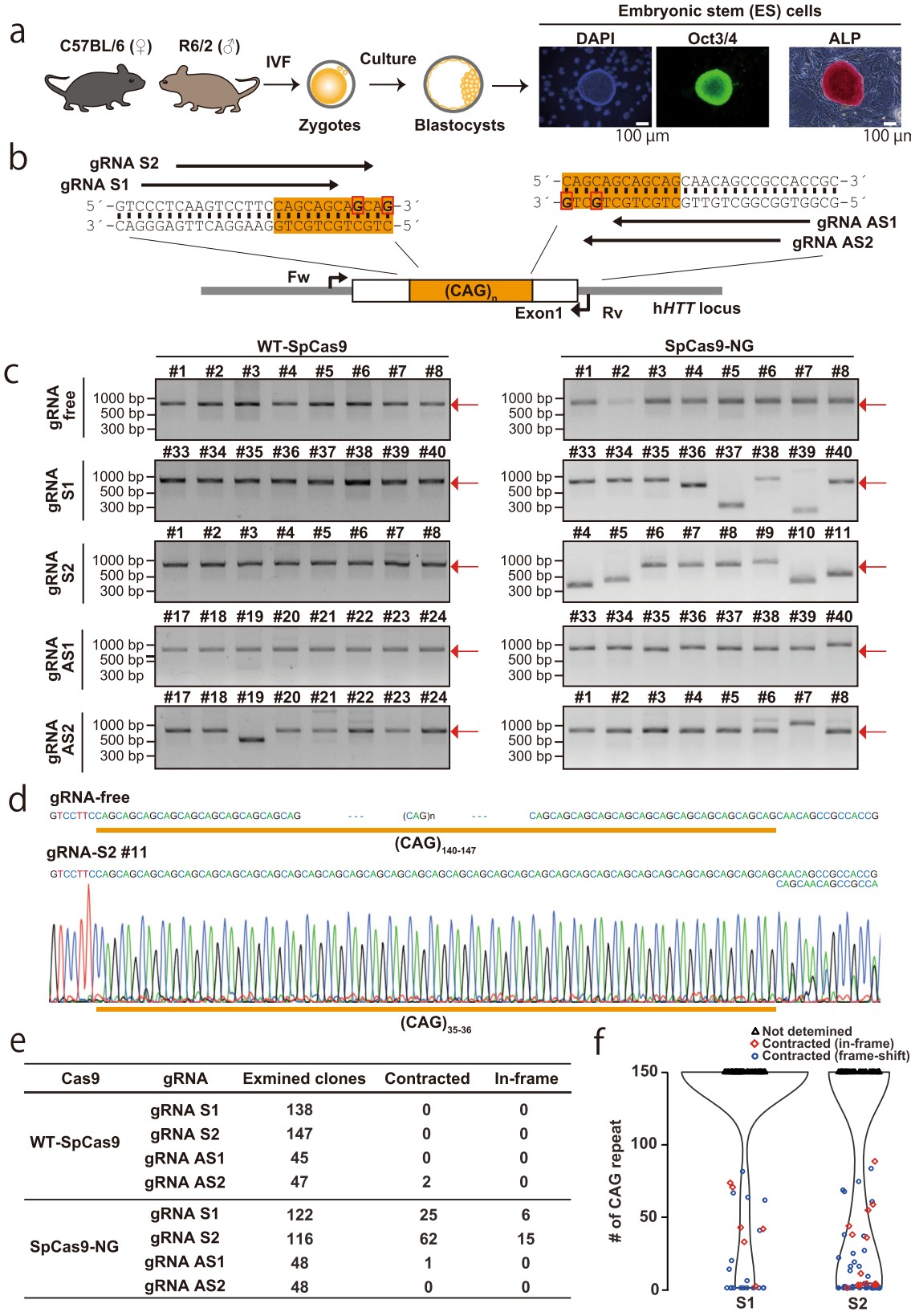

Many strategies have been explored for CRISPR/Cas9-mediated treatment for trinucleotide repeat disorders, including polyQ disorders[13–17,39]. The most straightforward application of CRISPR/Cas9 is to remove the entire CAG repeat tracts by designing two gRNAs outside the repeat sequence[13–16]. In contrast, we designed one gRNA on the boundary of the CAG repeat sequence using SpCas9-NG. Therefore, we could remove only repeat tracts and precisely repair the responsible genes. This is favorable to

**Fig. 2 CAG repeat contraction in R6/2 ES cells. a** Procedure for establishing ES cells from R6/2 mice. ES cell stemness was examined by Oct3/4 immunostaining and ALP activity. **b** Design of gRNAs and primers for genotyping (Fw and Rv). The orange background color shows the area of the repeat tract. Bold characters with a red box in the DNA sequence indicate the second G in NGN-PAMs. **c** PCR-based screening for successfully contracted clones. The red arrows indicate the original size of the PCR amplicon. **d** Direct sequencing of PCR amplicon. Clone #11 in gRNA-S2/SpCas9-NG are shown as an example. As we amplified human *HTT* transgene from mouse ES cells, the overlapping electrogram at the 3′ is due to a variety of repeat length or/and errors during PCR and sanger sequencing. **e** Summary of PCR-based screening and direct sequencing. **f** Violin plot for the number of CAG repeats in ES cell clones transfected with gRNA-S1/SpCas9-NG or gRNA-S2/SpCas9-NG expression vector. Black triangles indicate clones without detectable band-shift. Red diamonds and blue dots indicate clones with downward band-shift; red diamond: in-frame CAG repeat sequence; blue dot: out-of-frame CAG repeat sequence.

avoid unexpected side effects of removing surrounding sequences with gRNAs. In this regard, SpCas9-NG-mediated CAG repeat contraction can be an alternative to HD-based strategy[17] for the precise repair of expanded CAG repeat tract. The efficiencies of precise repair with gRNA-S1/SpCas9-NG and gRNA-S2/SpCas9-NG were 5% and 13% (Fig. 2e), respectively. Although these rates are not still satisfactory, they were comparable to or more efficient than the reported HD-based strategy (5%)[17]. More importantly, our strategy does not require donor DNA that may randomly integrate and destroy endogenous genes. Considering these advantages, SpCas9-NG-mediated CAG repeat contraction can be an alternative option for expanded CAG repeat repair.

We acknowledge there are several problems to be solved in our strategy. First, gRNAs we designed do not have specificity to pathogenic alleles, and therefore wild-type functional alleles might be disrupted upon genome editing. Therefore, the effect of reduced expression of the responsible genes has to be examined before applying our method. From previous reports, the reduction of *Htt* expression in mouse striatal neuronal cells did not affect viability[17]. The heterozygous mutant mice of *Atxn1/2/3*, also responsible for trinucleotide repeat disorders, are viable[40–42]. These results indicate that our strategy can be applied to *Htt* and *Atxn1/2/3* genes. Another problem to be solved is an unpredictable contraction of repeats; only two repeats remained in some clones after genome editing (Fig. 2d). In fact, a shorter CAG repeat in *Ar*, coding androgen receptor, than the normal range increases the risk of prostate cancer and hyperplasia due to higher sensitivity to sex hormones[43]. Considering that trinucleotide repeats might fulfill some physiological functions, a method to control the contraction has to be explored. One aspiring method is to shorten the longevity of Cas9 by ubiquitin tagging[44]. Another perspective method is to control the genome editing ability of Cas9 by splitting and tagging pMAG and nMAG (photoactivatable Cas9)[45].

In summary, we developed a gene-disruption-free CAG repeat contraction method based on SpCas9-NG, which has no risk of donor DNA random integration. Thus, SpCas9-NG can be a powerful tool for repairing abnormally expanded CAG repeats as well as other disease mutations that are difficult to access with WT-SpCas9. Considering that around 30% of the WT population ameliorated the phenotype of chimeric mice dramatically (Fig. 5d and e), induced neural stem cell transplantation or viral vector injection for SpCas9-NG delivery can be considered for translational research.

## Methods

**Animals and superovulation**. ICR (Japan SLC, Shizuoka, Japan), C57BL/6 (SLC), B6D2F1 (SLC), and R6/2 transgenic mice[19] were used. R6/2 mice carry the 5′ end of human *Htt* gene composed of the promoter and exon1 with expanded CAG repeats (140–147 repeats)[19]. For superovulation, pregnant mare serum gonadotropin (PMSG) (7.5 units, ASKA Pharmaceutical, Tokyo, Japan) or CARD HyperOva (KYUDO, Saga, Japan) was injected into the abdominal cavity of female mice, followed by human chorionic gonadotropin (hCG) (7.5 units, ASKA Pharmaceutical) 48 h after PMSG or HyperOva. All animal experiments were approved by the Animal Care and Use Committee of the Research Institute for Microbial Diseases, Osaka University, Japan (approval code: H30–01-0; approval date: 4 July 2018).

**Cell lines**. EGR-G01[46] ES cells were generated in Dr. Ikawa Lab and cultured in KnockOut DMEM (108297-018, Thermo Fisher Scientific) supplemented with 1% Penicillin-Streptomycin-Glutamine, 55 μM 2-mercaptoethanol, 1% Non-Essential Amino Acid Solution (11140-050, Thermo Fisher Scientific), 1% Sodium Pyruvate (11360-070, Thermo Fisher Scientific), 30 μM Adenosine (A4036, Sigma–Aldrich, St. Louis, MO, USA), 30 μM Guanosine (G6264, Sigma–Aldrich), 30 μM Cytidine (C4654, Sigma–Aldrich), 30 μM Uridine (U3003, Sigma–Aldrich), 10 μM Thymidine (T1895, Sigma–Aldrich), 100 U/ml mouse LIF, and 20% fetal bovine serum (FBS; 51650-500, Biowest, Nuaillé, France). HEK293T cell line was gifted from the Dr. Verma Lab and cultured in Dulbecco's Modified Eagle Medium (DMEM, High Glucose, Pyruvate; Thermo Fisher Scientific, Waltham, MA, USA) containing 10% FBS.

**Bacterial strains**. *Escherichia coli* (*E. coli*) strain DH5α (Toyobo, Osaka, Japan) was used for DNA cloning. *E. coli* cells were grown in LB or 2×YT medium containing 100 mg/L ampicillin and were transformed or cloned using standard methods.

**Vector construction for SpCas9 characterization and *Htt* targeting**. For genome editing in ESCs using puromycin, pX459-SpCas9-NG (Addgene plasmid #171370) was constructed by replacing the Cas9 coding sequence of pSpCas9(BB)-2A-Puro (PX459) V2.0 (Addgene plasmid #62988) with that of pX330-SpCas9-NG (Addgene plasmid #117919) using AgeI (R0552, NEB, MA, USA) and FseI (R0588, NEB) restriction enzymes. A DNA oligo (Genedesign, Osaka, Japan) was ligated into the BbsI (R0539, NEB) site of pX459-SpCas9-NG. Oligo DNA sequences are listed in Supplementary Data 1.

**Single-strand annealing (SSA) assay in HEK293T**. The SSA assay was performed as previously described[18] with slight modifications. Briefly, the 400–600 bp genomic fragments of *Rosa26*, *Cetn1*, and *Dnajb13* loci were inserted into MCS of pCAG-EGxxFP (Addgene plasmid #50716; for Cetn1 #50717). A mixture of 1 μg pCAG-EGFP-target and 1 μg pX459-SpCas9-NG with sgRNA sequences were introduced into $4 \times 10^5$ HEK293T cells/well in six-well-plates by the conventional calcium phosphate transfection method. The EGFP fluorescence was observed with a fluorescence microscope (BZ-X710, Keyence, Osaka, Japan) 36–48 h after transfection. The signal intensity was quantified using image J Fiji[47] (ver. 1.53c). Image processing protocols (.ijm file) and datasets are available from zenodo (https://doi.org/10.5281/zenodo.4768747). Briefly:

    run("Subtract Background…", "rolling = 10");

    run("Set Measurements…", "integrated redirect = None decimal = 3");

    run("Measure");

**Genome editing in ESCs and sample preparation for indel analysis**. For plasmid transfection, $1 \times 10^5$ EGR-G101 ESCs[46] were seeded on gelatin-coated six-well-plates with feeder cells. Mouse embryonic fibroblasts (MEF) were treated with 10 μg/mL mitomycin C (134-07991, WAKO, Osaka, Japan) and used as feeder cells. After 7–9 h, 1 μg pX459-SpCas9-NG with sgRNA sequences were introduced using Lipofectamine LTX & PLUS technology (15338-100, Thermo Fisher Scientific). After 14–18 h, the cells were selected with 1.0 μg/mL puromycin (ant-pr-1, InvivoGen, San Diego, CA, USA) for 48 h, grown for 3–4 more days without puromycin. Selected cells were then harvested and incubated in lysis buffer [20 mM Tris-HCl (pH 8.0), 5 mM EDTA, 400 mM NaCl, 0.3% SDS, and 200 μg/mL Actinase E solution] at 60 °C overnight.

**Genome editing in zygotes and sample preparation for indel analysis**. Superovulated B6D2F1 females were mated with B6D2F1 males, and eggs were collected from the oviduct (21 h after hCG injection). SpCas9 and SpCas9-NG were prepared as previously described[7] and incubated with synthesized crRNA (Sigma–Aldrich, St. Louis, MO, USA) and tracrRNA (#TRACRRNA05N-5NMOL, Sigma–Aldrich) at 37 °C for 5 min to make a gRNA/Cas9 ribonucleoprotein complex (40 ng/μL crRNA:tracrRNA, 100 ng/μL Cas9). crRNA and tracrRNA sequence are listed in Supplementary Data 1. The obtained complex was

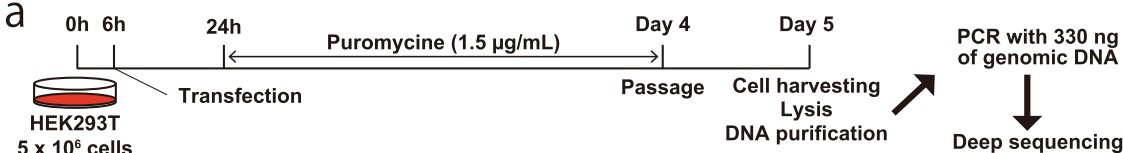

**Fig. 3 Off-target analysis of *HTT*-targeting gRNA. a** The procedure of off-target analysis in HEK293T cells. 330 ng of genomic DNA corresponding to $5.0 \times 10^4$ cells. **b** Off-target candidate sites and their mutation rate. The mutation rate was calculated by the following formula: [(total reads − no variants)/total reads]. For the on-target site, reads with in-frame 16–19 CAG repeats were summed up as no variants.

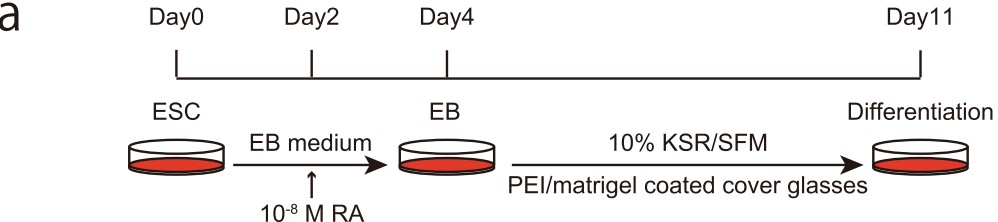

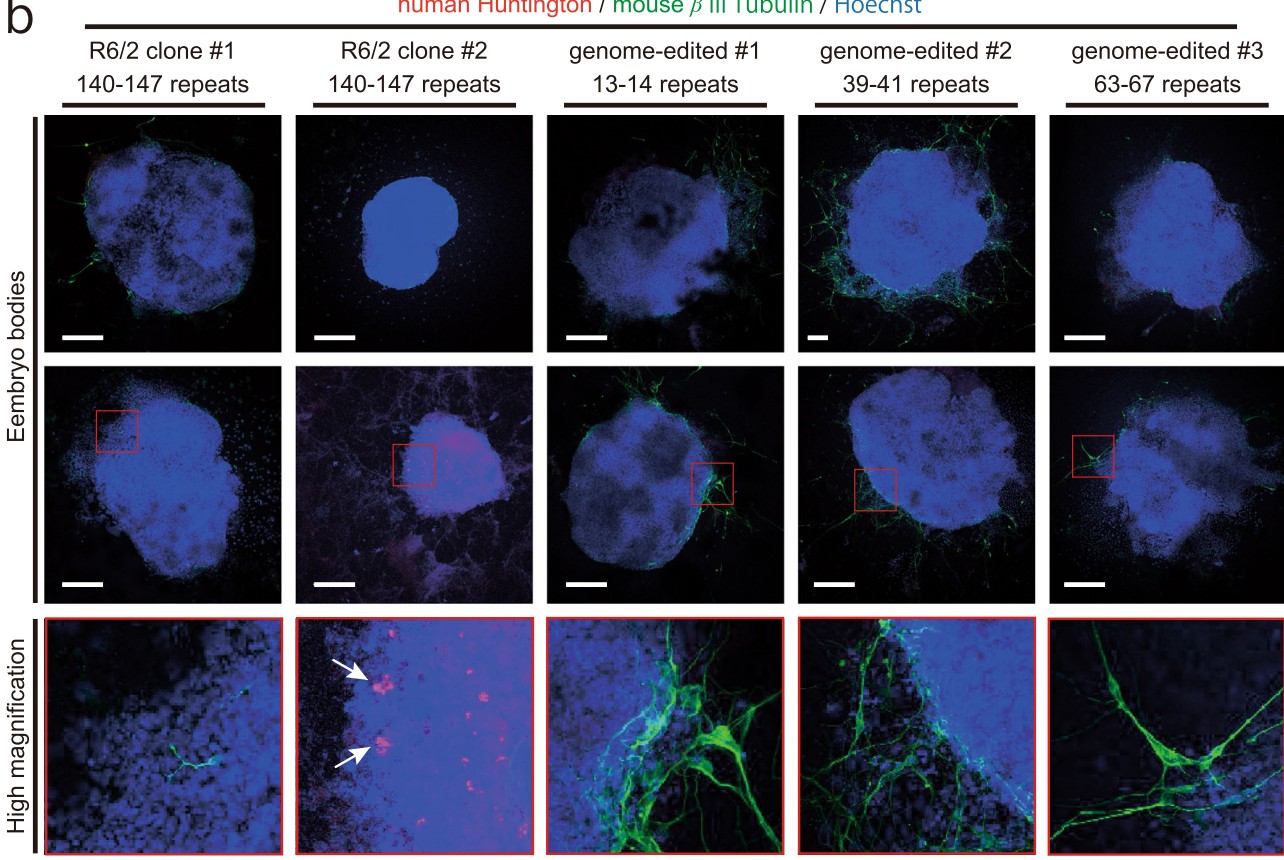

human Huntington / mouse β III Tubulin / Hoechst

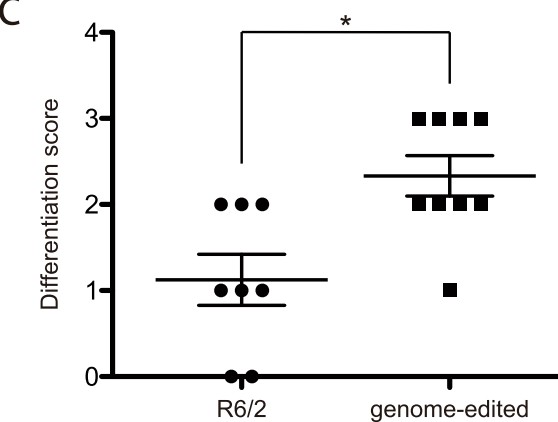

**Fig. 4 Differentiation of mock-treated and genome-edited R6/2 ES cells into neurons. a** The procedure of neuronal differentiation of ES cells. For in vitro differentiation assay, we used ES clones transfected with prototype SpCas9-NG (not published), not ES clones described in Fig. 2. **b** Immunostaining of differentiated embryoid bodies. White arrows in the high magnified image of R6/2 clone #2 indicate HTT aggregates. All scale bars indicate 100 μm: the EB in genome-edited #2 (top) was almost twice larger than the others. **c** The differentiation scores based on a percentage of the β III tubulin staining in the circumference of embryoid bodies; 1: 0–25%; 2: 25–50%; 3: 50–75%; 4: 75–100%. Statistical analysis was performed using a two-tailed Mann–Whitney U-test ($p$-value = 0.012).

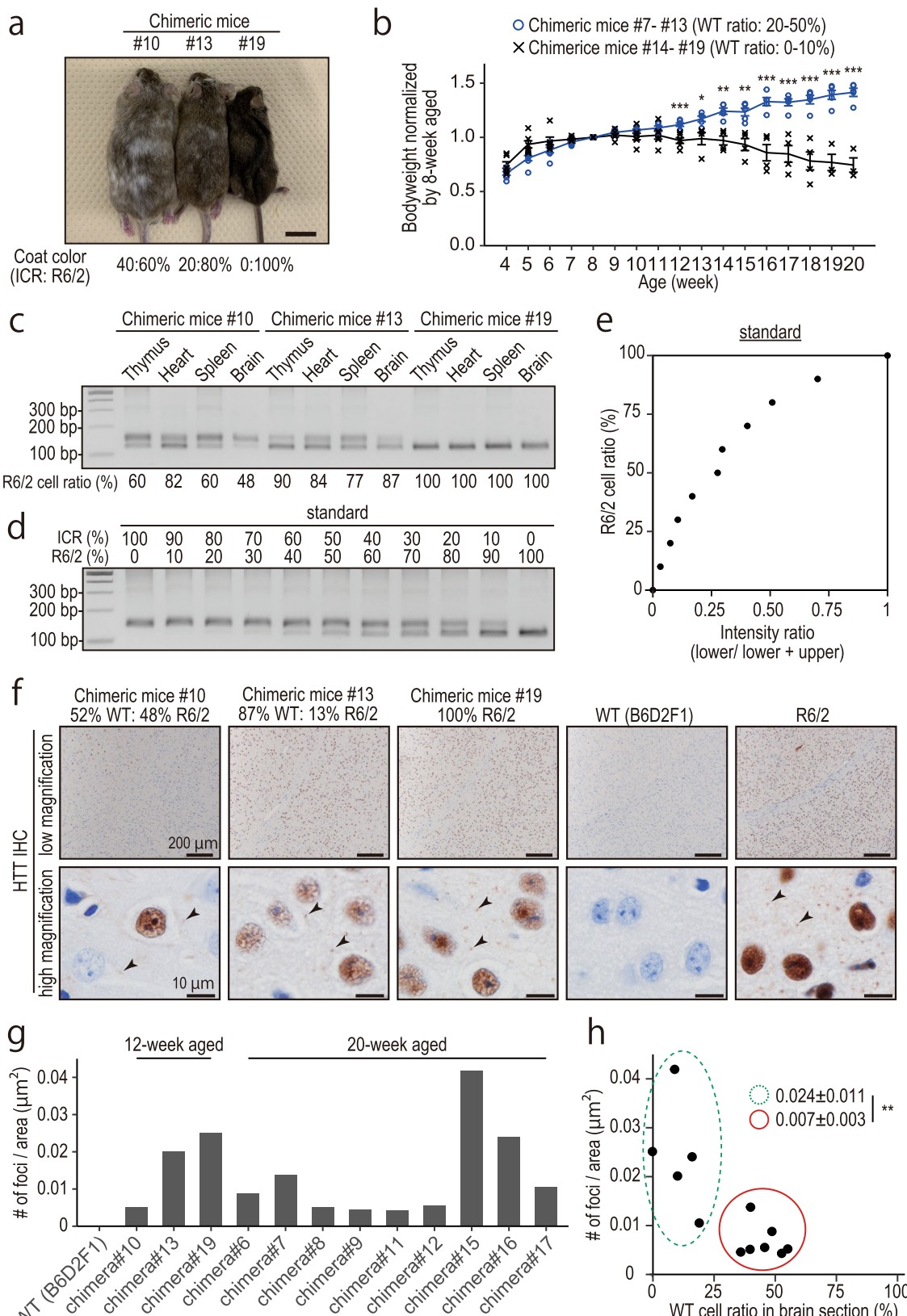

electroporated into zygotes using a super electroporator NEPA21 as previously described[48]. The eggs were cultivated in KSOM[49] for 4 days until they reached to the late blastocyst stage. Blastocysts were then collected in test tubes (8–10 eggs/ tube) and incubated in lysis buffer at 37 °C for 2 h.

**Indel analysis**. Using the pooled genomic samples prepared from ESCs and zygotes, polymerase chain reaction (PCR) with KOD FX neo (TOYOBO) was

performed. PCR products were purified using the Wizard SV Gel and PCR Clean-Up System (Promega, Madison, WI, USA) and Sanger sequenced. Primer sets used for PCR and sequencing are listed in Supplementary Data 1. Cleavage frequency was determined by TIDE software (https://tide.deskgen.com)[50].

**Derivation of ESC from R6/2 transgenic mice**. Oocytes were collected from the ampulla of each oviduct from superovulated C57BL/6 females (14 h after hCG

**Fig. 5 Phenotyping of R6/2-ESC->ICR chimeric mice. a** Chimeric mice and their chimeric ratio estimated by their coat color (ICR: white, R6/2-ESC: agouti). **b** Growth curve of chimeric mice. Chimeric mice were divided into two groups depending on their chimera ratio. Error bars indicate standard error. Statistical analysis was performed using a two-tailed student's $t$-test ($p$-value = 0.00089 [12 week], 0.0216 [13 week], 0.00128 [14 week], 0.00299 [14 week] <0.001 [~15 week]). **c–e** A microsatellite D15Mit266 was examined with PCR to determine the ESC contribution rate. The mixture of ICR and R6/2-ESC genomic DNA was used as the standard (**d**, **e**; approximate line: $y = 102x^3 - 262x^2 + 259x + 1.96$). Brain genomic DNA was extracted from the serial section examined in **f**. **f** Immunohistochemistry with anti-human HTT antibody. Black arrowheads indicate HTT aggregates. **g** The number of foci was calculated in randomly chosen three areas. **h** Relationship between the number of foci and R6/2 cell ratio in the brain. Each dot represents one animal. Green dashed circle and red circle indicate chimeric mice with less than 20% and more than 30% of WT cells, respectively. The error range (±) indicates standard deviation. Statistical analysis was performed using a two-tailed student's $t$-test ($p$-value = 0.0090).

injection) and incubated in CARD medium (KYUDO). Frozen-thawed R6/2 spermatozoa were incubated in FERTIUP medium for 30 min. The sperm suspension was added to CARD MEDIUM. After 3 hours of insemination, the eggs moved to KSOM medium until the blastocyst stage at 37 °C under 5% $CO_2$. Obtained blastocysts were then plated on a feeder layer in gelatin-coated dishes and cultured under 2i/LIF[51,52]. Eight days after plating, the expanded colonies derived from the inner cell mass (ICM) were obtained. Karyotyping analysis and sex determination were performed as previously described[51].

**Immunostaining and alkaline phosphatase detection in ES cells.** Immunostaining and alkaline phosphatase activity detection were performed as previously described[51]. Briefly, alkaline phosphatase (ALP) staining was carried out with a Histofine Fuchsin Substrate kit for ALP (415161, Nichirei, Tokyo, Japan), according to the manufacturer's instruction. Immunostaining was carried out as follows. First, ESCs were fixed with 4% paraformaldehyde (PFA; 166-23251, WAKO). After three washes with PBS, the fixed cells were stored overnight in PBS containing 1% goat serum and 0.1% Triton X-100 at 4 °C. The blocked cells were then incubated with primary antibody against OCT3/4 (PM048, MBL, Nagoya, Japan; 1:500) at 4 °C overnight. After three washes with PBS, the cells were incubated with Alexa Fluor 488 conjugated goat anti-rabbit IgG (R37116, Thermo Fisher Scientific; 1:200) at room temperature for 1 h. The cells were then washed three times with PBS and mounted with Vectashield mounting medium containing DAPI (H-1200, Vector Laboratories, Burlingame, CA, USA).

**CAG repeat contraction in R6/2 ESCs.** After transfecting ESCs with a plasmid expressing a gRNA/SpCas9-NG complex targeting *Htt* and selection with puromycin (see the section "Genome editing in ESCs and sample preparation for indel analysis"), the cells were dissociated with 0.25% trypsin (15090-046, Thermo Fisher Scientific), and seeded on gelatin-coated six-well plates ($1 \times 10^3$ cells) with feeder cells for cloning. The plated cells were picked up after 5–6 days, transferred onto gelatin-coated 96-well plates with feeder cells, and split in duplicate for freezing and DNA harvesting 48–72 h later. CAG repeat length was then examined by PCR amplification and direct sequencing. Primer sets used for PCR and sequencing are listed in Supplementary Data 1.

**Off-target analysis in HEK293T cells.** We transfected $5 \times 10^6$ HEK293T cells with 10 μg of gRNA-S1 or -S2/NG-Cas9/PuroR expressing vectors (pX459). After 18 h, the cells were selected with 1.5 μg/mL puromycin for 72 h, passaged and grown for 1 day without puromycin to remove dead cells. Selected cells were then harvested and incubated in lysis buffer [20 mM Tris-HCl (pH 8.0), 5 mM EDTA, 400 mM NaCl, 0.3% SDS, and 200 μg/mL Actinase E solution] at 60 °C overnight. We used 330 ng genomic DNA (corresponding to about $5 \times 10^4$ cells) for PCR amplification. Primer sets used for PCR are listed in Supplementary Data 1. After mixing the same amount of DNA fragments in each group (six groups: gRNA-S1/WT-SpCas9, gRNA-S1/SpCas9-NG, mock for gRNA-S1, gRNA-S2/WT-SpCas9, gRNA-S2/SpCas9-NG, mock for gRNA-S2), 100 ng of DNA was subjected to KAPA HyperPrep Kit (Roche) for DNA library preparation according to the manufacturer's instructions. Sequencing was performed on the NovaSeq6000 platform (Illumina) in a 151 bp pair-end mode. The NovaSeq6000 BCL2 format file was converted to FASTAQ format files using bcl2fastq2 ver.2.20 (Illumina), followed by adapter sequence removal by PEAT-1.2.4[53]. The sequence reads were mapped to the human reference genome sequence (hg38) using BWA-MEM algorithm[54], then converted to BAM format files. For off-target cleavage analysis, we used CrispRVariants-1.18.0[55].

**In vitro differentiation of R6/2 ESCs and immunostaining.** R6/2 ES cells ($2.5 \times 10^5$ cells) were cultured in 2 mL of embryoid body (EB) formation medium, ES medium without 2i/LIF, for 2 days. Cell aggregates were harvested and resuspended in EB formation medium supplemented with $10^{-8}$ M retinoic acid (R2625, Sigma) for 2 days. Single colonies were picked up and plated on polyethyleneimine (P3143, Sigma–Aldrich)/Matrigel Matrix (No. 356234, BD Biosciences)-coated cover glasses and cultured in 500 μL of 10% KSR (No. 10828028, Thermo Fisher Scientific)/ Serum-free media [SFM; DMEM/F12 (1:1), 5 mM HEPES buffer, 0.6% glucose, 3 mM $NaHCO_3$, 2 mM glutamine, 25 μg/ml insulin, 100 μg/ml transferrin,

20 nM progesterone, 60 μM putrescine, and 30 nM sodium selenite] for 7 days. The cells were fixed with 4% PFA in PBS at room temperature for 20 min and permeabilized with 0.3% Triton X-100 in PBS for 5 min. After washing with PBS, the fixed cells were blocked with PBS containing 10% goat serum and 0.05% Triton X-100 at room temperature for 1 h. The blocked cells were then incubated with an anti-huntingtin antibody (1:250; MAB5374, Millipore, Billerica, MA) at 4 °C overnight. After three washes with PBS, the cells were incubated with Alexa Fluor 555 conjugated goat anti-mouse IgG (A21422, Thermo Fisher Scientific; 1:4000) at room temperature for 2 h. The cells were then incubated with a mouse anti-β III tubulin mouse antibody (T8660, Merck; 1:1000) at 4 °C overnight. After three washes with PBS, the cells were incubated with Alexa Fluor 488 conjugated goat anti-mouse IgG (A11001, Thermo Fisher Scientific; 1:4000) and Hoechst 33258 (B2883, Sigma–Aldrich; 1 μg/ml) at room temperature for 2 h. The differentiation ability was determined by double-blinded scoring. Statical analysis and data presentation were performed using Prism 6 software (GraphPad Software Inc., La Jolla, CA, USA).

**Generation of chimeric mice.** The ESCs were injected into eight-cell ICR embryos, and the chimeric blastocysts were transplanted into the uteri of pseudopregnant females. After 17 days, pups were removed from the uteri and placed with foster mothers in the cage if the pups were not delivered naturally.

**Body weight and survival rate.** The body weight and survival were recorded weekly. Mice were housed together with their foster mothers up to 4 weeks after birth. After 4 weeks, males and females were separately housed.

**Immunohistochemistry of brain section.** Mice were deeply anesthetized with 0.3 mg/kg of medetomidine, 4.0 mg/kg of midazolam, and 5.0 mg/kg of butorphanol[56], and then intracardially perfused with 0.9% saline containing 10 μg/mL heparin, followed by 4% PFA in PBS (pH 7.4). Brains were removed and post-fixed in 4% PFA in PBS at 4 °C. Fixed brains were dehydrated by increasing concentrations of ethanol and xylene and embedded with paraffin. Rehydrated paraffin sections (5 μm) were incubated in 3% $H_2O_2$ at room temperature for 5 min for endogenous peroxidase inactivation and blocked and permeabilized in PBS containing 3% horse serum and 0.1% Triton X-100 for 30 min at room temperature. The sections were then incubated with a mouse anti-huntingtin antibody (1:250) at 4 °C overnight. After three washes with PBS, the sections were incubated with goat anti-rat IgG-micropolymer HRP (MP-7402, Vector Laboratories, Burlingame, CA, USA) for 1 hour. The sections were then washed three times with PBS and incubated in ImmPACT DAB (SK-4105, Vector Laboratories, Burlingame, CA, USA) working solution for 1 min, counterstained with Mayer's hematoxylin solution for 3 min, dehydrated in increasing ethanol concentrations, and finally mounted with Permount (SP15-100-1, Ferma, Tokyo, Japan). The sections were observed using a BX53 (olympus, Tokyo, Japan) microscope. The number of foci was counted using image J Fiji[47] (ver. 1.53c). Image processing protocols (.ijm file) and datasets are available from zenodo (https://doi.org/10.5281/zenodo.4768747). Briefly:

    run("Colour Deconvolution", "vectors = [H DAB] hide");

    run("Threshold…");

    setThreshold(0, 200);

    setOption("BlackBackground", true);

    run("Convert to Mask");

    run("Analyze Particles…", "size = 0.10–100.00 show = Outlines exclude clear include summarize");

**Immunofluorescence of brain section.** Rehydrated paraffin sections (5 μm) were boiled in pH 6.0 sodium citrate buffer for 10 min, blocked and permeabilized with 10% goat serum and 0.1% Triton X-100 for 30 min in PBS, and incubated with a

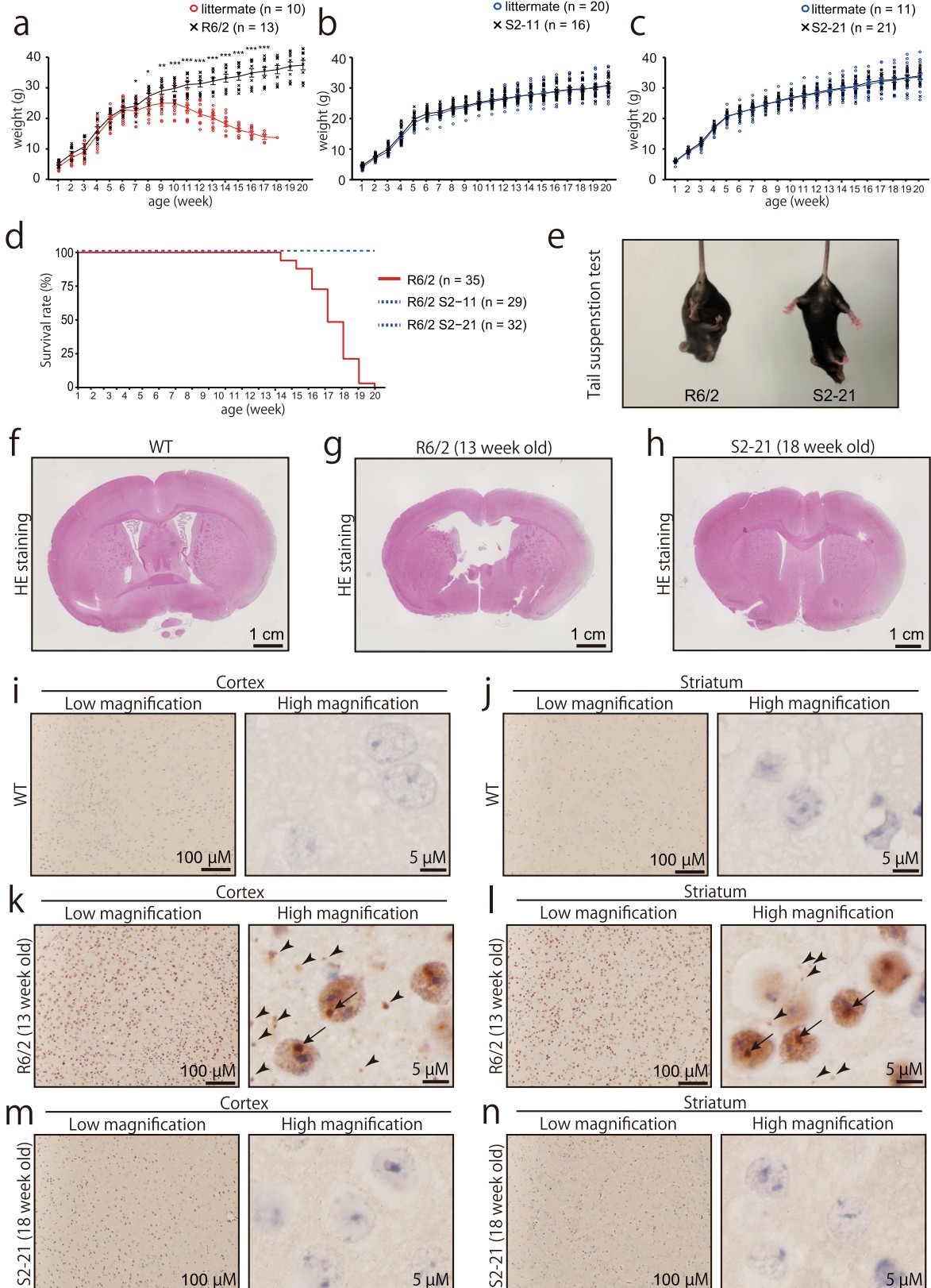

**Fig. 6 Phenotypic analysis of R6/2 and isogenic mice produced from repeat-corrected ES cells. a–c** Growth curve of R6/2 (**a**) and genome-edited (**b**, **c**) mice. Male mice were used for plotting and data analysis. Statistical analysis was performed using a two-tailed student's *t*-test (*p*-value = 0.033 [8 week], 0.018 [9 week], 0.004 [10 week], <0.001 [~11 week]). **d** Survival curve of R6/2 and genome-edited mice. **e** Tail suspension test of R6/2 (left) and isogenic mice (right). See supplemental video 1. **f–h** Histological analysis with H&E staining of R6/2 (**g**; 13 week old) and isogenic mice (**h**; 18 week old). Isogenic mice were analyzed at the age when almost all R6/2 mice were dead. **i–n** Immunohistochemistry with anti-human HTT antibody. Black arrows and arrowheads indicate HTT aggregates inside and outside the nucleus, respectively.

**Table 1 Body weight of chimeric mice was normalized by body weight at 8 week old for comparison.**

| Chimeric mice (Chimeric ratio; ICR: R6/2) | | Body weight (normalized by 8 week aged) | | | | | | | | | | | | |
|---|---|---|---|---|---|---|---|---|---|---|---|---|---|---|
| | | Weeks of age | | | | | | | | | | | | |
| | | 8 | 9 | 10 | 11 | 12 | 13 | 14 | 15 | 16 | 17 | 18 | 19 | 20 |
| ICR | | 1.0 | 1.0 | 1.1 | 1.1 | 1.2 | 1.3 | 1.4 | 1.3 | 1.4 | 1.4 | 1.5 | 1.6 | 1.6 |
| ICR | | 1.0 | 1.0 | 1.0 | 1.0 | 1.0 | 1.0 | 1.1 | 1.1 | 1.1 | 1.1 | 1.0 | 1.1 | 1.1 |
| ICR | | 1.0 | 1.1 | 1.1 | 1.2 | 1.2 | 1.2 | 1.3 | 1.3 | 1.3 | 1.3 | 1.5 | 1.4 | 1.4 |
| 1 | (90%: 10%) | 1.0 | 1.1 | 1.1 | 1.2 | 1.2 | 1.3 | 1.4 | 1.4 | 1.5 | 1.5 | 1.6 | 1.6 | 1.6 |
| 2 | (90%: 10%) | 1.0 | 1.0 | 1.0 | 1.1 | 1.2 | 1.3 | 1.4 | 1.3 | 1.5 | 1.4 | 1.6 | 1.5 | 1.6 |
| 3 | (90%: 10%) | 1.0 | 1.2 | 1.1 | 1.2 | 1.4 | 1.4 | 1.4 | 1.5 | 1.7 | 1.5 | 1.6 | 1.7 | 1.7 |
| 4 | (90%: 10%) | 1.0 | 1.1 | 1.1 | 1.1 | 1.1 | 1.1 | 1.1 | 1.1 | 1.2 | 1.1 | 1.2 | 1.3 | 1.3 |
| 5 | (70%: 30%) | 1.0 | 1.1 | 1.1 | 1.3 | 1.2 | 1.4 | 1.3 | 1.5 | 1.5 | 1.6 | 1.5 | 1.6 | 1.7 |
| 6 | (70%: 30%) | 1.0 | 1.0 | 1.1 | 1.1 | 1.1 | 1.2 | 1.2 | 1.2 | 1.2 | 1.2 | 1.2 | 1.3 | 1.3 |
| 7 | (50%: 50%) | 1.0 | 1.0 | 1.1 | 1.1 | 1.1 | 1.1 | 1.1 | 1.2 | 1.2 | 1.2 | 1.2 | 1.2 | 1.3 |
| 8 | (50%: 50%) | 1.0 | 1.0 | 1.0 | 1.0 | 1.2 | 1.1 | 1.2 | 1.1 | 1.3 | 1.3 | 1.3 | 1.4 | 1.4 |
| 9 | (50%: 50%) | 1.0 | 1.0 | 1.1 | 1.1 | 1.1 | 1.2 | 1.2 | 1.3 | 1.3 | 1.3 | 1.4 | 1.4 | 1.4 |
| 10 | (40%: 60%) | 1.0 | 1.0 | 1.1 | 1.1 | 1.1 | * | | | | | | | |
| 11 | (40%: 60%) | 1.0 | 1.1 | 1.1 | 1.1 | 1.2 | 1.2 | 1.3 | 1.3 | 1.3 | 1.4 | 1.4 | 1.5 | 1.5 |
| 12 | (30%: 70%) | 1.0 | 1.1 | 1.1 | 1.1 | 1.1 | 1.2 | 1.3 | 1.3 | 1.4 | 1.4 | 1.4 | 1.5 | 1.5 |
| 13 | (20%: 80%) | 1.0 | 1.0 | 1.0 | 1.1 | 1.1 | * | | | | | | | |
| 14 | (10%: 90%) | 1.0 | 1.0 | 1.0 | 1.0 | 0.9 | 0.8 | 0.9 | 0.8 | 0.7 | 0.7 | 0.6 | X | |
| 15 | (10%: 90%) | 1.0 | 1.0 | 1.0 | 1.0 | 1.0 | 1.1 | 0.9 | 0.9 | 0.7 | 0.8 | 0.7 | 0.7 | 0.6 |
| 16 | (10%: 90%) | 1.0 | 1.0 | 1.0 | 1.1 | 1.0 | 1.1 | 1.0 | 1.0 | 1.0 | 1.0 | 1.0 | 0.9 | 0.9 |
| 17 | (10%: 90%) | 1.0 | 1.0 | 1.0 | 1.0 | 1.0 | 1.0 | 1.1 | 1.0 | 1.0 | 0.9 | 0.9 | 0.7 | 0.7 |
| 18 | (0: 100%) | 1.0 | 1.1 | 1.1 | 1.2 | X | | | | | | | | |
| 19 | (0: 100%) | 1.0 | 1.0 | 0.9 | 0.9 | 0.9 | * | | | | | | | |

The chimeric ratio was estimated from their coat color. Chimeric mice #10, #13, and #19 (*) were used for sampling at 12 weeks of age. Chimeric mice #14 and #18 were dead at 11 and 19 weeks of age, respectively.

mouse anti-huntingtin antibody (1:250) overnight at 4 °C or 1 h. After three washes with PBS, the sections were incubated with Alexa Fluor 488 conjugated goat anti-mouse IgG (A32723, Thermo Fisher Scientific; 1:200) at room temperature for 1 h. The sections were then washed three times with PBS and mounted with Vectashield mounting medium containing DAPI. The sections were observed using a BX53 (olympus) microscope.

**H&E staining of brain section.** Rehydrated paraffin sections (5 μm) were stained with Mayer hematoxylin solution for 5 min, counterstained with eosin Y solution [53% (v/v) ethanol, 0.3% (v/v) eosin, and 0.5% (v/v) acetic acid] for 2 min, dehydrated in increasing ethanol concentrations, cleared in xylene, and finally mounted in Entellan new (Merck, Kenilworth, NJ, USA). The sections were observed using a BX53 (olympus) microscope.

**Statistics and reproducibility.** Statistical analysis was performed using a two-tailed Mann–Whitney U-test and two-tailed student's $t$-test ($*p \le 0.05$, $**p \le 0.01$, and $***p \le 0.001$) using R and Prism 6 (GraphPad Software Inc.), respectively. The meaning of error bars was clarified in each legend (standard error or standard deviation). The sample numbers were described in each legend.

**Reporting summary.** Further information on research design is available in the Nature Research Reporting Summary linked to this article.

## Data availability
Sanger sequences for genome-edited R6/2-ESC clones and full imaging data supporting the conclusion of this paper are available from zenodo (https://doi.org/10.5281/zenodo.4768747)[57]. Fastq data for off-target analysis were deposited to Sequence Read Archive (SRA; PRJNA727674). Source data underlying the graphs and charts are also available as Supplementary Data 2–6. Any remaining information can be obtained from the corresponding author upon reasonable request.

## Code availability
The data were plotted graphically by using ggplot2[58] (Figs. 1, 2, 5, and 6) and Prism 6 (GraphPad Software Inc.; Fig. 4). For image analysis and cropping, we used image J Fiji[47] (ver. 1.53c). For indel analysis, we used TIDE software (https://tide.deskgen.com)[50]. For the processing of NGS data and off-target analysis, we used bcl2fastq2 ver.2.20 (Illumina), PEAT[53], BWA-MEM[54], and CrispRVariants-1.18.0[55]. Image J macros (.ijm files), R scripts (.R files), and datasets for analysis are available from zenodo (https://doi.org/10.5281/zenodo.4768747)[57].

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

## Acknowledgements
We would like to thank Dr Julio Castaneda for the critical reading of the manuscript. This work was supported by: the Japan Society for the Promotion of Science (JSPS) KAKENHI grants (JP19J21619 to S.O., JP18K14612 and JP20H03172 to T.N., 26291010 and 15H01463 to H.N., and JP19H05750 and 21H04753 to M.I.); Japan Agency for Medical Research and Development (AMED) grants (JP20am0401005h0002 to H.N. and O.N., JP19ek0109222 to Y.N., and JP20gm5010001 to M.I.); Takeda Science Foundation Grants to T.N. and M.I.; The Nakajima Foundation to T.N.; and the Bill & Melinda Gates Foundation (INV-001902 to M.I.).

## Author contributions
S.O., T.N., and M.I. conceived the study. S.O., H.N., O.N., and Y.N. prepared materials. S.O. and T.N. performed genome editing experiments and analyzed the data. S.O. performed CAG repeat contraction experiment and analyzed the data. S.O. performed an off-target experiment and analyzed the data. M.N. and S.H. performed a neural differentiation experiment and analyzed the data. S.O. performed mouse experiments and analyzed the data. S.O. and M.I. wrote the paper.

## Competing interests
The authors declare no competing interests.
