## [Transparent Peer Review File · Communications Biology]

Reviewers' comments:

Reviewer #1 (Remarks to the Author):

In this manuscript, Oura et al compared wild type Cas9 (WT-SpCas9) that requires an NGG PAM and a modified Cas9 (SpCas9-NG) that recognizes NGN PAM. NGN PAM is particularly useful for targeting CAG repeats-containing genes. The authors found that Sp-Cas9-NG is indeed able to target the HD gene, which has expanded CAG repeats, in ES cells from R6/2 mice. They also generated new R6/2 mouse line by targeting CAG repeats in ES cells.

The authors were exploring an important issue, as effective targeting CAG-containing gene has a great therapeutic potential. However, they did not perform rigorous experiments to demonstrate that Sp-Cas9-NG is superior to WT-SpCas9 in eliminating expanded CAG repeats.

First, they used sequencing of PCR clones to evaluate targeting efficiency, which can be largely influenced by cloning and PCR. The authors should perform deep sequencing to have more quantitative analysis.

Second, Sp-Cas9-NG is supposed to have more off-targets than WT-Cas9. The authors need to perform whole genome sequencing to compare with WT-Cas9 for off-target events.

Third, Sp-Cas9-NG's effect appears to be cell-type dependent, because it worked better in ES cells but not in zygotes. The authors should test its effect in the primary neuronal cells. The analysis of new R6/2 line with different CAG repeats do not critically contribute to the current paper. Instead, the authors should consider direct targeting the brain of R6/2 mice to see if Sp-Cas9-NG can yield any therapeutic outcome.

Reviewer #2 (Remarks to the Author):

Oura et al. report a CRISPR-based strategy to reduce the CAG repeat expansion associated with Huntington Disease. In a transgenic mouse model, the authors use CRISPR to cut at the edges of the HTT CAG-repeat tract in order to stimulate contraction. Particularly, they use SpCas9-NG, a variant with increased activity at NGN PAMs, to target the NGC PAMs of the repeat tract in mouse ES cells and zygotes. This is a bold strategy that contrasts other published examples which aimed to simply knockout HD alleles (cited works), and could provide a permanent solution in contrast to targeting RNA with sustained Cas expression, as shown recently for myotonic dystrophy (doi.org/10.1038/s41551-020-00607-7). Still, WT or even repaired alleles are at risk for re-cleavage, which the authors admit in the discussion. The authors also present data contrasting the activities of WT SpCas9 and SpCas9-NG in mouse ES cells and zygotes. Curiously, the activities of WT-SpCas9 and SpCas9-NG differ on degenerate PAM sequences between ES cells and zygotes. Finally, they show a reversal of HD phenotypes in mice derived from gene edited ES cells.

Overall, the manuscript is intriguing and concisely presented. The phenotypic reversion is made clear through quantitative and visual (movie) data. However, there are some concerns about the overarching premise. Presumably, SpCas9-NG is critical for their strategy, However, if we follow the logic and data of the paper, it seems that the mice used in phenotypic modelling are derived from ES cells edited in-frame with WT-SpCas9 and one gRNA (Figure 2e). While this is irrelevant for phenotyping the gene correction, it raises concerns over WT-SpCas9 promiscuity, and questions the necessity for SpCas9-NG to achieve their goal.

Major concerns

The authors need to clarify the source of the cell lines used for phenotyping. In Figure 2c (one hit method), in-frame repair was only seen with WT-SpCas9. The two-hit method was successful using SpCas9-NG (Supplementary Figure 2b). Were ES clones S2-11 and S2-21 derived using SpCas9-NG or WT-SpCas9? This is critical to support the title, statements in the abstract, and overall premise of the paper.

The strategy requires positioning gRNAs precisely, and in some cases gRNAs overlap CAG repeats. Yet the authors do not assay or calculate off-target cleavage potential. How specific are the gRNAs in mouse? human? For example, RGEN Tools Cas-OFFinder shows that AS2 has 14 human and 17 mouse off-target sites with only one mismatch, and hundreds with 2 mismatches. Moreover, off-target potential is expected to increase by relaxing PAM requirements. The authors should address the potential of their gRNAs for off-target cleavage, and discuss approaches to attenuate it.

Two alternative strategies (citations 15, 16) used nickases to reduce the frequency of expansion. Although the authors observed mainly contraction, this strategy may help avoid potential off-target indel formation. This point should at least be discussed.

The authors claim applications in "gene therapy", but how do they envision it to be applied? Have the authors tested the approach in human cells? (note that HEK293T cells are used in Figure 1).

Minor concerns

Line 18 – "The most common trinucleotide sequence is CAG..."

Line 20 – "A common symptom of polyQ disorders..."

References are required.

Line39: "appearing after homology directed repair (HDR)"

Line49: "Consistent with the result of SSA assay above"

Was the assay SSA or HDR?

Figure 1b, how was the GFP intensity calculated?

Figure 2b or c, the PCR amplicon size should be indicated.

Figure 2c and Supplementary Figure 2a, the lanes should be labeled.

Figure 3b, What are clones #1-3? Staining for Human Huntington is difficult to see. Do the two rows represent two EBs? The scale bar in genome-edited #2 (top) is shorter than the others.

Figure 4 a-c, control littermates are included. In e-h, normal brain sections could be included for comparison.

Figure 1b, spCas9 > SpCas9; 1c, Lirpofection > Lipofection; 1c/e, analysiss > analysis, equence > sequence

Figure 4d legend R6/2 2-11 and R6/2 2-21 should be S2-11 and S2-21 (?)

Supplementary Figure 3 s2-11 should be S2-11

Reviewer #3 (Remarks to the Author):

Oura et al., utilized SpCas9-NG, a variant of SpCas9 with flexible PAM sequence, to excise CAG repeat within the transgenic human HTT gene locus in mouse ES cells derived from R6/2 mice. Interestingly, single gRNA (one-hit) was sufficient to delete the CAG repeat, compared with dual gRNA (two-hit) approach. They performed neural differentiation of the genome edited mouse ES cells and found that the deletion of the CAG repeat enhances the Tubb3+ neuronal differentiation capacity. When mice were generated from the two clones of the repeat-deleted mouse ES cells, body weight, lifespan, feet-clasping posture, tremors, cerebral atrophy and Htt aggregation were reverted, as expected.

The use of SpCas9-NG for HD's repeat is an interesting approach, however, technical advancements over the related papers are not clear. Overall, this could be an interesting manuscript for readers in the genome editing field, but several technical questions need to be addressed. The reviewer's specific comments are below.

1) Superiority upon the published works

The related papers (i.e. ref. 15-20) should be explained in the introduction. Compared with the previous genome editing approaches, such as double nicking, what is the advantage to use SpCas9-NG? Ideally, this should be experimentally demonstrated side-by-side.

2) Repeat deletion strategy

The rate of in-frame clones was 21% (9/43 clones) for "one-hit" (Fig. 2e) and 24% (6/25) for two-hit" (Fig. S2b). Is this enough for correcting neurons in HD patient? How comparable with other CRISPR approaches? Also, statistical analysis should be performed between the one-hit and two-hit groups.

3) ESC differentiation phenotype

Huntington's disease patient develop normally until adult, but start developing symptoms around their 30's in general. This is true for R6/2 mouse model and embryonic development is rather normal, until the age of 5-6 weeks. However, when the authors performed neuronal differentiation from mouse ES cells for 11 days, impaired differentiation phenotype was observed. What kinds of neurons and which stage of neurons generated with this differentiation protocol, and which phenotype in the HD patient is recapitulated with this cellular model? How this is comparable to the HD-iPSC model reported previously (i.e. ref. 19)?

4) Phenotype analysis of chimeric mice

There is no surprise at all if a mouse is generated from a ES cell line after deletion of the pathogenic CAG repeat.

For the mouse assays in Fig.4, the chimeric rate should be described. In the gene therapy/genome editing therapy field, what percentage of cells should be genetically corrected to improve phenotypes is an important question, and this can be partly addressed by analyzing the chimeric mice.

The chimeric rate is also important if the genome edited ES cells did not survive or selectively depleted in the ICR embryo. Disease control mice should be chimeric mice established by injection of uncorrected R6/2 ES cells.

5) A mechanism of the large deletion by just one sgRNA

P18: "This large deletion occurred, probably because CAG repeat tracts become unstable slipped-strand structures upon double-strand breaks^{13,14}."

To proof this, more sgRNA target should be tested, like directly targeting within the CAG repeat.

6) Target specificity of SpCas9-NG

One major challenge associated with targeting CAG repeat is the abundance of the sequence in the

genome. In addition, target specificity of the PAM-relaxed version of SpCas9-NG is not demonstrated in this manuscript. At least, several related sequence site in the mouse genome should be investigated to validate the specificity of the sgRNAs used in this study.

<Minor points>

Fig.1a, b: This is not directly related with the context of the manuscript, so the reviewer recommend to move to a Supplementary Figure.

Fig.1d and Fig.1f: Despite the same sgRNA was used, It is interesting to see the Dnajp13 gRNAs with NGA and NGC PAM did not work for zygote genome editing. Is this true if the same plasmid DNA was used, instead of RNP electroporation?

P19: "the average cleavage efficiencies of SpCas9-NG at NGA/NGT/NGC sites were comparable to those of WT-SpCas9"

This is not a fair comparison, as NGA and NGC PAM did not work well, whereas NGT PAM did.

Fig.2b: PAM sequence should be clearly marked.

Fig.2c: Size marker should be provided. What was the biggest size of deletion and expansion?

Fig.2d: The overlapping electrogram at the 3' end should be explained in the figure legend, if it is a heterogenous pattern. Subcloning with E.coli and Sanger sequencing to distinguishing each allele is recommended.

Fig.2e: The label of WT and Cas9-NG is most likely inverse.

Fig. S2: This should be a part of main Figure 2.

Fig.3b: The number of Htt aggregation spot should be quantified.
Please provide higher magnification image, as human HTT spot and Tubb3 signal is not visible.

Fig. 3f,g and 3h,i: Quantitative analysis, along side with the correlation with the chimeric rate, should be performed.

Response to the reviewers:

Reviewer #1

In this manuscript, Oura et al compared wild type Cas9 (WT-SpCas9) that requires an NGG PAM and a modified Cas9 (SpCas9-NG) that recognizes NGN PAM. NGN PAM is particularly useful for targeting CAG repeats-containing genes. The authors found that Sp-Cas9-NG is indeed able to target the HD gene, which has expanded CAG repeats, in ES cells from R6/2 mice. They also generated new R6/2 mouse line by targeting CAG repeats in ES cells.

The authors were exploring an important issue, as effective targeting CAG-containing gene has a great therapeutic potential. However, they did not perform rigorous experiments to demonstrate that Sp-Cas9-NG is superior to WT-SpCas9 in eliminating expanded CAG repeats.

First, they used sequencing of PCR clones to evaluate targeting efficiency, which can be largely influenced by cloning and PCR. The authors should perform deep sequencing to have more quantitative analysis.

- We thank the reviewer for the valuable advice. We agree with the necessity of quantitative analysis. However, next-generation sequencers tend to read shorter fragments preferentially, and 450 bp (150 CAG repeat) is still beyond their capacity. Therefore, instead of deep sequencing, we examined additional 96 ES cell clones to make the data more reliable and made a semi-quantitative violin plot graph (Fig. 2F).

Second, Sp-Cas9-NG is supposed to have more off-targets than WT-Cas9. The authors need to perform whole genome sequencing to compare with WT-Cas9 for off-target events.

- Whole genome sequencing (WGS) can be an unbiased genome-wide method for identifying off-target events. However, considering WGS is difficult to detect infrequent mutations, we performed PCR-seq for expected off-target sites (Fig. 3). We added sentences regarding off-target events in **L. 140-150**, and **L.231-236**.

L.140-150: "To examine off-target events in human cells, we transfected HEK293T cells with gRNA-S1 and -S2 expression vector (Fig. 3a), and examined mutation rate in all candidate sites by PCR-seq. The off-target candidate sites have the exact match of seed sequence (12mer) and more than 5 other nucleotides. With gRNA-S1, SpCas9-NG efficiently cleaved the on-target site than WT-SpCas9 (Fig.3b and Supplemental Fig. 3a) while mutation rates at off-target sites were comparable even with mock-transfected groups. On the other hand, we detected higher off-target mutations with gRNA-S2, although gRNA-S2/SpCas9-NG was more efficient in the on-target site than gRNA-S1/SpCas9-NG (Fig.3b and Supplemental Fig. 3). These result suggest that the gRNA-S1 has a lower risk in human HTT targeting."

L.231-236: "As the off-target analysis shows, our strategy is a trade-off between efficient CAG repeat

contraction and low off-target cleavages. Although genome-wide off-target analysis²⁵⁻²⁷ in human neural ES/iPS cells is ideal, our result showed gRNA-S1 has less off-target risk than gRNA-S2. High fidelity mutations²⁸⁻³⁴ are needed to be introduced into SpCas9-NG to minimize the risk. It should be noted that the use of Cas9 nickase (D10A or N863A mutations) also has been reported to reduce off-target event³⁵⁻³⁷. “

Third, Sp-Cas9-NG's effect appears to be cell-type dependent, because it worked better in ES cells but not in zygous. The authors should test its effect in the primary neuronal cells. The analysis of new R6/2 line with different CAG repeats do not critically contribute to the current paper. Instead, the authors should consider direct targeting the the brain of R6/2 mice to see if Sp-Cas9-NG can yield any therapeutic outcome.

- We agreed that whether Sp-Cas9-NG can contract the CAG repeat in primary neurons and the brain is critical for applying our method to gene therapy. We hopefully tackle the point in future research, but these application studies are beyond the focus of the current paper presenting the efficacy of CAS9-NG to target the repetitive CAG. Instead, we envisioned the therapeutic application of our strategy in the discussion, **L.272-274**.

L. 272-274: “Considering that 30% of the normal population ameliorated the phenotype of chimeric mice (Fig. 5d and e), induced neural stem cell transplantation or viral vector injection for SpCas9-NG delivery can be considered for translational research.”

Reviewer #2

Oura et al. report a CRISPR-based strategy to reduce the CAG repeat expansion associated with Huntington Disease. In a transgenic mouse model, the authors use CRISPR to cut at the edges of the HTT CAG-repeat tract in order to stimulate contraction. Particularly, they use SpCas9-NG, a variant with increased activity at NGN PAMs, to target the NGC PAMs of the repeat tract in mouse ES cells and zygotes. This is a bold strategy that contrasts other published examples which aimed to simply knockout HD alleles (cited works), and could provide a permanent solution in contrast to targeting RNA with sustained Cas expression, as shown recently for myotonic dystrophy (doi.org/10.1038/s41551-020-00607-7). Still, WT or even repaired alleles are at risk for re-cleavage, which the authors admit in the discussion. The authors also present data contrasting the activities of WT SpCas9 and SpCas9-NG in mouse ES cells and zygotes. Curiously, the activities of WT-SpCas9 and SpCas9-NG differ on degenerate PAM sequences between ES cells and zygotes. Finally, they show a reversal of HD phenotypes in mice derived from gene edited ES cells.

Overall, the manuscript is intriguing and concisely presented. The phenotypic reversion is

made clear through quantitative and visual (movie) data. However, there are some concerns about the overarching premise. Presumably, SpCas9-NG is critical for their strategy, However, if we follow the logic and data of the paper, it seems that the mice used in phenotypic modelling are derived from ES cells edited in-frame with WT-SpCas9 and one gRNA (Figure 2e). While this is irrelevant for phenotyping the gene correction, it raises concerns over WT-SpCas9 promiscuity, and questions the necessity for SpCas9-NG to achieve their goal.

Major concerns

The authors need to clarify the source of the cell lines used for phenotyping. In Figure 2c (one hit method), in-frame repair was only seen with WT-SpCas9. The two-hit method was successful using SpCas9-NG (Supplementary Figure 2b). Were ES clones S2-11 and S2-21 derived using SpCas9-NG or WT-SpCas9? This is critical to support the title, statements in the abstract, and overall premise of the paper.

- We are sorry for the confusion. S2-11 and S2-21 are ES cell clones transfected with the SpCas9-NG expression vector. We revised Cas9 labels column in Fig. 2E.

The strategy requires positioning gRNAs precisely, and in some cases gRNAs overlap CAG repeats. Yet the authors do not assay or calculate off-target cleavage potential. How specific are the gRNAs in mouse? human? For example, RGEN Tools Cas-OFFinder shows that AS2 has 14 human and 17 mouse off-target sites with only one mismatch, and hundreds with 2 mismatches. Moreover, off-target potential is expected to increase by relaxing PAM requirements. The authors should address the potential of their gRNAs for off-target cleavage, and discuss approaches to attenuate it.

- We thank the reviewer for clarifying a critical point. As AS1 and AS2 did not cleave the on-target sites, we examined several potential off-target sites for S1 and S2 gRNA by PCR-seq using HEK cells (Fig. 3). We included the results and discussions in **L. 140-150**, and **L.231-236**, respectively.

L.140-150: "To examine off-target events in human cells, we transfected HEK293T cells with gRNA-S1 and -S2 expression vector (Fig. 3a), and examined mutation rate in all candidate sites by PCR-seq. The off-target candidate sites have the exact match of seed sequence (12mer) and more than 5 other nucleotides. With gRNA-S1, SpCas9-NG efficiently cleaved the on-target site than WT-SpCas9 (Fig.3b and Supplemental Fig. 3a) while mutation rates at off-target sites were comparable even with mock-transfected groups. On the other hand, we detected higher off-target mutations with gRNA-S2, although gRNA-S2/SpCas9-NG was more efficient in the on-target site than gRNA-S1/SpCas9-NG (Fig.3b and Supplemental Fig. 3). These result suggest that the gRNA-S1 has a lower risk in human HTT targeting."

L.231-236: "As the off-target analysis shows, our strategy is a trade-off between efficient CAG repeat

contraction and low off-target cleavages. Although genome-wide off-target analysis²⁵⁻²⁷ in human neural ES/iPS cells is ideal, our result showed gRNA-S1 has less off-target risk than gRNA-S2. High fidelity mutations²⁸⁻³⁴ are needed to be introduced into SpCas9-NG to minimize the risk. It should be noted that the use of Cas9 nickase (D10A or N863A mutations) also has been reported to reduce off-target event³⁵⁻³⁷. “

Two alternative strategies (citations 15, 16) used nickases to reduce the frequency of expansion. Although the authors observed mainly contraction, this strategy may help avoid potential off-target indel formation. This point should at least be discussed.

- Thank you for pointing this out. We agree that SpCas9-NG-nickases can be a great future consideration to reduce off-target cleavages. We added the sentences regarding the usefulness of nickases in the discussion (**L. 235-236**)

L. 235-236: “It should be noted that the use of Cas9 nickase (D10A or N863A mutations) also has been reported to reduce off-target event³⁵⁻³⁷ “

The authors claim applications in “gene therapy”, but how do they envision it to be applied? Have the authors tested the approach in human cells? (note that HEK293T cells are used in Figure 1).

- We envisioned neural stem cell transplantation or viral vector injection into the brain, although we did not try these experiments in this paper. However, at least, we confirmed CAG repeat contraction in human cells (HEK293T), as a critical implication for gene therapy. We showed the result in **Fig. S3** and mentioned our envision for gene therapy in **L.272-274**.

L. 272-274: “Considering that 30% of the normal population ameliorated the phenotype of chimeric mice (Fig. 5d and e), induced neural stem cell transplantation or viral vector injection for SpCas9-NG delivery can be considered for translational research.”

Minor concerns

Line 18 – “The most common trinucleotide sequence is CAG...”

Line 20 – “A common symptom of polyQ disorders...”

References are required.

- We change the sentence and add references as following: One of the most common trinucleotide sequence implicated in human diseases is CAG^{8,9}.
- We added two references^{10,11} in Line 20.

8 Mirkin, S. M. Expandable DNA repeats and human disease. Nature 447, 932-940, doi:10.1038/nature05977 (2007).

9 Schmidt, M. H. M. & Pearson, C. E. Disease-associated repeat instability and mismatch repair. *DNA repair* **38**, 117-126, doi:10.1016/j.dnarep.2015.11.008 (2016).

10 Paulson, H. L., Shakkottai, V. G., Clark, H. B. & Orr, H. T. Polyglutamine spinocerebellar ataxias - from genes to potential treatments. *Nature reviews. Neuroscience* **18**, 613-626, doi:10.1038/nrn.2017.92 (2017).

11 Stoyas, C. A. & La Spada, A. R. The CAG-polyglutamine repeat diseases: a clinical, molecular, genetic, and pathophysiologic nosology. *Handbook of clinical neurology* **147**, 143-170, doi:10.1016/b978-0-444-63233-3.00011-7 (2018).

Line39: "appearing after homology directed repair (HDR)"

Line49: "Consistent with the result of SSA assay above"

Was the assay SSA or HDR?

- Homology directed repair (HDR) include homologous recombination (HR) or single strand annealing (SSA) in its definition. It's difficult to know which occurred in the cells (HR vs. SSA), although this kind of gRNA assay is often called SSA assay. To avoid confusing readers, we removed the word SSA from the sentence.

Figure 1b, how was the GFP intensity calculated?

- We used ImageJ software to calculate GFP intensity. We added image processing protocols in the material method section (**L.306-308**).

Figure 2b or c, the PCR amplicon size should be indicated.

- We indicated amplicon size.

Figure 2c and Supplementary Figure 2a, the lanes should be labeled.

- We labeled the lanes.

Figure 3b, What are clones #1-3? Staining for Human Huntington is difficult to see. Do the two rows represent two EBs? The scale bar in genome-edited #2 (top) is shorter than the others.

- We did *in vitro* differentiation assay with ES clones transfected with prototype SpCas9-NG (not published), not with ES cell clones described in Figure 2. We clarified this point in the figure legend: "For in vitro differentiation assay, we used ES clones transfected with prototype SpCas9-NG (not published), not ES clones described in Figure 2."
- We rarely see human Huntington staining, probably due to reduced differentiation ability or apoptotic elimination.

- Two rows represent two EBs. We added labels on the left side.
- The EB in genome-edited #2 (top) was almost twice larger than the others. We clarified this point in the figure legend.

Figure 4 a-c, control littermates are included. In e-h, normal brain sections could be included for comparison.

- Thank you for pointing this out. We added normal brain sections and rearranged **Figure 6** and **S4** (Figure 4 and S3 in original).

Figure 1b, spCas9 > SpCas9; 1c, Lirpofection > Lipofection; 1c/e, analysiss > analysis, equence > sequence

Figure 4d legend R6/2 2-11 and R6/2 2-21 should be S2-11 and S2-21 (?)

Supplementary Figure 3 s2-11 should be S2-11

- Thank you for the note. We revised all mistakes.

Reviewer #3

Oura et al., utilized SpCas9-NG, a variant of SpCas9 with flexible PAM sequence, to excise CAG repeat within the transgenic human HTT gene locus in mouse ES cells derived from R6/2 mice. Interestingly, single gRNA (one-hit) was sufficient to delete the CAG repeat, compared with dual gRNA (two-hit) approach. They performed neural differentiation of the genome edited mouse ES cells and found that the deletion of the CAG repeat enhances the Tubb3+ neuronal differentiation capacity. When mice were generated from the two clones of the repeat-deleted mouse ES cells, body weight, lifespan, feet-clasping posture, tremors, cerebral atrophy and Htt aggregation were reverted, as expected.

The use of SpCas9-NG for HD's repeat is an interesting approach, however, technical advancements over the related papers are not clear. Overall, this could be an interesting manuscript for readers in the genome editing field, but several technical questions need to be addressed. The reviewer's specific comments are below.

1) Superiority upon the published works

The related papers (i.e. ref. 15-20) should be explained in the introduction. Compared with the previous genome editing approaches, such as double nicking, what is the advantage to use SpCas9-NG? Ideally, this should be experimentally demonstrated side-by-side.

- We thank the reviewer for pointing this out. We also explained the related papers in the introduction section (**L.72-L.78**).

L.72-L.78: "CRISPR/Cas9 can be a breakthrough for causal treatment by directly targeting and repairing responsible genes. The most straightforward application of CRISPR/Cas9 is to excise the expanded repeat tracts¹³⁻¹⁶. This was accomplished by designing two gRNAs outside the repeat tracts, disrupting the CAG repeats and extra intragenic sequences. An alternative strategy is to integrate exogenous DNA into the loci (homologous recombination; HR) to precisely repair the responsible genes¹⁷. However, HR efficiency is lower than NHEJ."

- We showed the advantages of our strategy more clearly in the discussion section (**L.237-L.251**) to suggest that our strategy could be one option for CAG repeat repair. Although Magdalena *et al.* reported a double nicking paper entitled "Precise Excision of the CAG Tract from the Huntingtin Gene by Cas9 Nickases", they removed CAG repeat with surrounding sequences. Therefore, we categorized this report in the same group with other Cas9-based excision strategies.

L.237-L.251: "Many strategies have been explored for CRISPR/Cas9-mediated treatment for trinucleotide repeat disorders, including polyQ disorders^{13-17,38}. The most straightforward application of CRISPR/Cas9 is to remove the entire CAG repeat tracts by designing two gRNAs outside the repeat sequence¹³⁻¹⁶. In contrast, we designed one gRNA on the boundary of the CAG repeat sequence using SpCas9-NG. Therefore, we could remove only repeat tracts and precisely repair the responsible genes. This is favorable to avoid unexpected side effects of removing surrounding sequences with gRNAs. In this regard, SpCas9-NG-mediated CAG repeat contraction can be an alternative to HD-based strategy¹⁷ for the precise repair of expanded CAG repeat tract. The efficiency of precise repair with gRNA-S1 and -S2/SpCas9-NG was 5% and 13% (Fig. 2e), respectively. Although these rates are not still satisfactory, they were comparable to or more efficient than the reported HD-based strategy (5%)¹⁷. More importantly, our strategy does not require donor DNA that may randomly integrate and destroy endogenous genes. Considering these advantages, SpCas9-NG-mediated CAG repeat contraction can be an alternative option for expanded CAG repeat repair."

2) Repeat deletion strategy

The rate of in-frame clones was 21% (9/43 clones) for "one-hit" (Fig. 2e) and 24% (6/25) for two-hit" (Fig. S2b). Is this enough for correcting neurons in HD patient? How comparable with other CRISPR approaches? Also, statistical analysis should be performed between the one-hit and two-hit groups.

- From the chimeric analysis, around 30% of normal neurons can alleviate the symptom (**Fig. 5d and 5e**), indicating 21% of repair is still helpful. However, we admit that the alleviation might be due to the difference of genetic background of mouse strains.
- We mentioned the efficiency of other approaches in the discussion section.

- The gRNAs AS1 and AS2 showed almost no cleavage activity on *HTT* target. This means we could not carry out the two-hit strategy. To avoid readers' confusion, we changed sentences in the discussion section (**L.227 - L.230**).

L.227-L.230: "We could not evaluate the two-hit method in this study because gRNA-AS1 and -AS2 could not guide Cas9 to the target sequence efficiently (Fig. 2c and e). More trials on different targets with different gRNA sets will be needed to conclude which method is better."

3) ESC differentiation phenotype

Huntington's disease patient develop normally until adult, but start developing symptoms around their 30's in general. This is true for R6/2 mouse model and embryonic development is rather normal, until the age of 5-6 weeks. However, when the authors performed neuronal differentiation from mouse ES cells for 11 days, impaired differentiation phenotype was observed. What kinds of neurons and which stage of neurons generated with this differentiation protocol, and which phenotype in the HD patient is recapitulated with this cellular model? How this is comparable to the HD-iPSC model reported previously (i.e. ref. 19)?

- The differentiated neural cells expressed β III tubulin, an early neural marker. On the other hand, mature neural cell markers (Map2, NeuN) were negative. Therefore, most of the cells probably pre-differentiated default GABAergic neurons. We clarified that β III tubulin is an early neural marker in **L.156**.
- As the reviewer noticed, the HD patients/mice show no neural differentiation phenotype during development. Further, as explained later, previous reports showed HD hiPSCs were vulnerable to apoptosis. Therefore, in our cellular model, neural cell maintenance might be defective. However, we only examined neural cell existence and cannot conclude the cause of reduced neural cell number. To avoid exaggeration, we changed the sentences (**L.157-158**).

L.157-L.158: "Fig. 4b and 4c, we observed more neural cells in genome-edited R6/2 embryo bodies (EBs) than in mock-treated R6/2 EBs."

- Xu *et al.* did not detect any mutant HTT-containing aggregates in neurons differentiated from HD hiPSCs. HD iPSC Consortium [1] did not see the aggregate (data not shown) in immunocytochemistry, although they detected mutant HTT in neural stem cells in immunoblot analysis. Consistent with these previous reports, we rarely see mutant HTT aggregates.

Regarding low neural cell numbers, HD iPSC Consortium reported a gradual dying off of differentiated neurons over time. Further, both groups reported massive apoptosis upon BDNF withdrawal, suggesting HD iPSC is susceptible to cell death. It has been

reported that BDNF promotes neural cell survival, and BDNF production decreases in HD patients and in HD iPSC-derived neural cells. We did not supplement BDNF during neural differentiation. This might explain the dramatic decrease of differentiated neural cells.

[1] HD iPSC Consortium. Induced pluripotent stem cells from patients with Huntington's disease show CAG-repeat-expansion-associated phenotypes. *Cell Stem Cell*. 2012 Aug 3;11(2):264-78.

4) Phenotype analysis of chimeric mice

There is no surprise at all if a mouse is generated from a ES cell line after deletion of the pathogenic CAG repeat. For the mouse assays in Fig.4, the chimeric rate should be described. In the gene therapy/genome editing therapy field, what percentage of cells should be genetically corrected to improve phenotypes is an important question, and this can be partly addressed by analyzing the chimeric mice. The chimeric rate is also important if the genome edited ES cells did not survive or selectively depleted in the ICR embryo. Disease control mice should be chimeric mice established by injection of uncorrected R6/2 ES cells.

- We used F1 mice obtained from R6/2 ES chimeric mice in Fig.5 (Fig.4 in original), because the phenotype highly depends on chimeric rate, as the reviewer notices. We clarify the point in the result section (**L. 182-184**).
L.182-L.184: "To confirm the CAG repeat contraction reverse the phenotype completely, we mated s2-11 and s2-21 chimeric mice with WT C57BL/6N mice to produce F1 mice and analyzed their phenotype."
- To answer the review comment regarding the relationship between the percentage of genetically corrected cells and phenotype severity, we first tried to make chimeric mice by injecting genome-edited R6/2 ES cells into R6/2 embryos. However, we could not obtain enough pups, probably because of the low quality of blastocysts (a small number of oocyte donor females, inefficient IVF with froze-thaw R6/2 sperm, and 4 days in vitro culture). Next, we injected R6/2 ES cells into WT ICR embryos and obtained a considerable number of pups. As the result, 20-30% of WT cells in host R6/2 mice alleviated the weight reduction. However, this alleviation might be due to the difference of genetic background.
- We compared chimeric rate in various organs and could not observe a low contribution rate of uncorrected R6/E2 cells in the brain, indicating that R6/2 ES cell-derived neuron did not undergo elimination (**Fig. 5c**).
- The HTT aggregation index decreased exponentially corresponding to reduction of R6/2 ES cell contribution (i.e. increase of WT population) (**Fig. 5d and e**). Notably, 50% of WT cells in brain showed sevenfold decreases in the HTT aggregation index (Chimera#11: 1.5 ± 0.4 ; Chimera#19: 10.8 ± 1.1 ; **Fig. 5e**). This result indicates that

repairing abnormally expanded CAG repeats in small population dramatically improve the symptom.

- We described these result in **L. 163-179**.

5) A mechanism of the large deletion by just one sgRNA

P18: "This large deletion occurred, probably because CAG repeat tracts become unstable slipped-strand structures upon double-strand breaks^{13,14}."

To proof this, more sgRNA target should be tested, like directly targeting within the CAG repeat.

- As suggested, we directly targeted the inside of the CAG repeat. However, the treated cells could not survive, probably due to extremely high DNA double-strand break occurrence (mouse genomic DNA possesses 10,587 off-target sites with 20mer matches). Instead, we targeted other endogenous CAG repeat sequences in mouse ES cells and observed repeat contraction, indicating that our strategy can be applied to any CAG/CTG repeat sequences. We described these results in **L.129-138** and in Figure. **S2d** and **S2e**.

L.129-138: "To examine one-hit method is efficient enough for CAG repeat contraction, we targeted the 3 longest endogenous CAG repeat sequences in mouse genomic DNA (chr7: 36,559,029–36,559,121 [31 repeats]; chr13: 4,490,789–4,490,884 [32 repeats]; and chr17: 55,547,368–55,547,460 [31 repeats]). We designed gRNAs in the same way with gRNA-S1 and gRNA-AS1, using the 3rd–4th CAG sequence as PAMs (Supplementary Fig. 2c). Consistent with the result of HTT CAG repeat, we observed downsized band shift more efficiently in SpCas9-NG than in WT-SpCas9 (Supplementary Fig. 2d and 2e). All gRNA-ASs showed less targeting efficiency than gRNA-Ss. These results indicate that one gRNAs designed on the boundary are enough to remove long CAG repeat tracts.

6) Target specificity of SpCas9-NG

One major challenge associated with targeting CAG repeat is the abundance of the sequence in the genome. In addition, target specificity of the PAM-relaxed version of SpCas9-NG is not demonstrated in this manuscript. At least, several related sequence site in the mouse genome should be investigated to validate the specificity of the sgRNAs used in this study.

- We again thank the reviewer for pointing out an important issue. As the off-target analysis in mouse cells is not informative, we transfected human cells (HEK293T) with S1 and S2 gRNA expression vectors and examined 28 potential off-target sites with the exact match of seed sequence (12mer) and more than 5 other nucleotides. As the result, gRNA-S1 did not cleave these off-target site, indicating gRNA-S1 has less off-target risk. We described these results in **L.141-148** and in **Figure. 3**.

L.141-148: "To examine off-target events in human cells, we transfected HEK293T cells with gRNA-S1 and -S2 expression vector (Fig. 3a), and examined mutation rate in all candidate sites by PCR-seq (Fig.

3b). The off-target candidate sites have the exact match of seed sequence (12mer) and more than 5 other nucleotides. With gRNA-S1, SpCas9-NG efficiently cleaved on-target site than WT-SpCas9 while mutation rates at off-target sites were comparable even with mock-transfected groups. On the other hand, we detected higher off-target mutations with gRNA-S2. These results suggest that the gRNA-S1 has a lower risk in human HTT targeting."

<Minor points>

Fig.1a, b: This is not directly related with the context of the manuscript, so the reviewer recommends to move to a Supplementary Figure.

➤ Thank you for the suggestion. However, we would like to keep Fig. 1a and 1b in the main figure because arranging side-by-side makes it easy to compare genome editing efficiency.

Fig.1d and Fig.1f: Despite the same sgRNA was used, it is interesting to see the Dnajp13 gRNAs with NGA and NGC PAM did not work for zygote genome editing. Is this true if the same plasmid DNA was used, instead of RNP electroporation?

➤ We really thank the reviewer for pointing out this. We tried plasmid DNA injection and observed improved cleavage efficiency in SpCas9-NG compared with WT-SpCas9. We added the result in Fig.1 and mentioned the result in **L.110-112**, and changed several sentences in the discussion section (**L.208-214**)

L.110-112: "On the other hand, when microinjecting gRNA/SpCas9 expressing plasmid into zygotes, SpCas9-NG cleaved the 3 out of 3 target sequences more efficiently than WT-SpCas9 (Fig. 1f)."

L.208-214: "As a note, the efficiency of gRNA/SpCas9-NG ribonucleoprotein complex-mediated genome editing was not satisfactory in our hand. Previously, Fujii et al. successfully generated a tyrosinase knockout by microinjecting SpCas9-NG mRNA and gRNA into the cytoplasm of zygotes²¹. The long-expression might be needed for SpCas9-NG due to its reduced cleavage efficiency (see NGG sites of Fig.1b, d, and f), as we only see efficient cleavage with plasmid injection."

P19: "the average cleavage efficiencies of SpCas9-NG at NGA/NGT/NGC sites were comparable to those of WT-SpCas9" This is not a fair comparison, as NGA and NGC PAM did not work well, whereas NGT PAM did.

➤ We removed those sentences.

Fig.2b: PAM sequence should be clearly marked.

➤ We emphasized 2ndG of PAM sequences.

Fig.2c: Size marker should be provided. What was the biggest size of deletion and expansion?

➤ We indicated amplicon size.

➤ The biggest size of downward band shift we could detect was 500 — 600 bp, which is

beyond whole length of 150 CAG repeat tract (450bp). The biggest size of upward band shift we could detect was 50 — 100 bp, corresponding to 17 — 33 repeat expansion.

Fig.2d: The overlapping electrogram at the 3' end should be explained in the figure legend, if it is a heterogenous pattern. Subcloning with E.coli and Sanger sequencing to distinguishing each allele is recommended.

- As we amplified human *HTT* transgene from mouse ES cells, the overlapping electrogram at the 3' is due to variety of repeat length or/and errors during PCR and sanger sequencing. We explained it in the figure legend.

Fig.2e: The label of WT and Cas9-NG is most likely inverse.

- Thank you for pointing out the mistake. We exchanged the label.

Fig. S2: This should be a part of main Figure 2.

- As explained “2) Repeat deletion strategy”, we could not carry out two-hit strategy. Although we should have removed that part from our manuscript to avoid confusing readers, we have decided to present that data as supplementary figure to show the concept of two-hit strategy.

Fig.3b: The number of Htt aggregation spot should be quantified. Please provide higher magnification image, as human HTT spot and Tubb3 signal is not visible.

- We could not see and count human Huntington staining in most of EBs, probably due to reduced differentiation ability or apoptotic elimination. We provide high magnification images.

Fig. 3f,g and 3h,i: Quantitative analysis, along side with the correlation with the chimeric rate, should be performed.

- As explained in “4) Phenotype analysis of chimeric mice”, we used F1 mice in Figure 6 (original Figure 4). Instead of quantitative analysis in Figure 6, we counted the number of foci in chimeric mice produced in revise experiments (**Fig.5**).

REVIEWERS' COMMENTS:

Reviewer #2 (Remarks to the Author):

The authors have made exceptional effort to provide additional data and revise the manuscript according to the suggestions of the reviewers. Apart from some minor spelling errors in the new figure labels (Fig3b 'Mutaion', Fig4b 'Eembryo', etc.), I have no further concerns and recommend the article for publication in Communications Biology.

Reviewer #3 (Remarks to the Author):

In the revised manuscript by Oura S et al., the authors have performed additional analyses of chimeric mice to show necessary contribution rate of CAG-repeat contracted cells, and potential off-target sites were assessed to investigate the risk of their gRNAs (S1 and S2) in human cultured cell context. Now, major concerns were adequately addressed. The reviewer has only a few comments in regards to the revised part as below.

Fig.3: "To examine off-target events in human cells, we transfected HEK293T cells with gRNA-S1 and -S2 expression vector (Fig. 3a)," Why the authors thought to test their mouse Htt targeting gRNAs in human cells? On-target sequence was identical between mouse Htt and human HTT? Better explanation is needed to avoid confusion for readers.

Fig.5d: There are two "WT" H&E panels. What is the difference between the two panels? Both necessary? Why the number of HTT aggregation spots looks so different (right panel should be Huntington model)? Also, Color legend should be included to show which dot represent Htt staining in the Figure legend.

Fig.5e: Why Chimera mice #11, #13, and #19 were analyzed here, instead of #10, #12, #18 as indicated in Fig. 5c and 5d? Especially, what is the contribution rate of Chimera #19 mouse? The chimeric contribution rate should be indicated in the graph or figure legend. Also, what the "BDF1" mean?

Fig.5a: The authors mentioned that "Chimeric mice with 30% or more WT cells kept gaining weight (Fig. 5a), whereas those with less than 10% of WT cells lost weight after 12 weeks of age, indicating around 30% of genetically corrected cells can alleviate the symptom." However, the cut-off of the "30% chimeric contribution" on body weight is not clear from the Fig. 5a. Are there any statistical difference between >30% and <10% groups?

To claim the necessary chimeric contribution rate to ameliorate a Huntington phenotype, not only the body weight, but also Htt aggregation index should be taking into account. It would be nice to discuss the similarity and difference from previous chimeric mouse studies.

Reply to Reviewers

In the revised manuscript by Ours S et al., the authors have performed additional analyses of chimeric mice to show necessary contribution rate of CAG-repeat contracted cells, and potential off-target sites were assessed to investigate the risk of their gRNAs (S1 and S2) in human cultured cell context. Now, major concerns were adequately addressed. The reviewer has only a few comments in regards to the revised part as below.

Fig.3: "To examine off-target events in human cells, we transfected HEK293T cells with gRNA-S1 and -S2 expression vector (Fig. 3a)," Why the authors thought to test their mouse Htt targeting gRNAs in human cells? On-target sequence was identical between mouse Htt and human HTT? Better explanation is needed to avoid confusion for readers.

Sorry for the confusion. R6/2 mice have human HTT exon1 as a transgene. To avoid confusing readers, we added the phrase "human HTT" to Line 130-131.

L.130-131: "Then, we designed four gRNAs on the boundary of human HTT CAG repeat tracts"

Fig.5d: There are two "WT" H&E panels. What is the difference between the two panels? Both necessary? Why the number of HTT aggregation spots looks so different (right panel should be Huntington model)? Also, Color legend should be included to show which dot represent Htt staining in the Figure legend.

Thank you for pointing this out. We changed the label of right panel to R6/2. To show HTT aggregates we counted, we also showed high magnified images.

Fig.5e: Why Chimera mice #11, #13, and #19 were analyzed here, instead of #10, #12, #18 as indicated in Fig. 5c and 5d? Especially, what is the contribution rate of Chimera #19 mouse? The chimeric contribution rate should be indicated in the graph or figure legend. Also, what the "BDF1" mean?

We thank the reviewer again for pointing out our mistakes. We corrected the label of chimeric mice. BDF1 means a hybrid mouse (B6D2F1). We corrected BDF1 to B6D2F1 throughout the manuscript.

Fig.5a: The authors mentioned that "Chimeric mice with 30% or more WT cells kept gaining weight (Fig. 5a), whereas those with less than 10% of WT cells lost weight after 12 weeks of age, indicating around 30% of genetically corrected cells can alleviate the symptom." However, the cut-off of the "30% chimeric contribution" on body weight is not clear from the Fig. 5a. Are there any statistical difference between >30% and <10% groups?

We kept measuring weight of chimeric mice until 20 weeks of age and the 20% threshold was supported and they were statistically different between >20% and <10% groups (Figure 5b).

To claim the necessary chimeric contribution rate to ameliorate a Huntington phenotype, not only the body weight, but also Htt aggregation index should be taking into account. It would be nice to discuss the similarity and difference from previous chimeric mouse studies.

We also examined HTT aggregation index of chimeric mice at 20 weeks of age and added these results to Figure 5g-h. We also discussed the similarity to previous chimeric analysis (Line-183-185). L.183-185: "This result indicated around 20–30% of genetically corrected cells could alleviate the symptom, which agrees with the previous chimeric study indicating 30–70% of WT cell contribution delayed the onset of R6/2 symptom"²⁰